# PGC1/PPAR drive cardiomyocyte maturation at single cell level via YAP1 and SF3B2

Sean A. Murphy[1,2,3], Matthew Miyamoto [1,2,3], Anaïs Kervadec [4], Suraj Kannan[1,2,3], Emmanouil Tampakakis[1], Sandeep Kambhampati [1,2,3], Brian Leei Lin [1], Sam Paek[5], Peter Andersen[1], Dong-Ik Lee [1], Renjun Zhu[1,2,3], Steven S. An [5], David A. Kass [1], Hideki Uosaki [1,3,6], Alexandre R. Colas [4] & Chulan Kwon [1,2,3✉]

Cardiomyocytes undergo significant structural and functional changes after birth, and these fundamental processes are essential for the heart to pump blood to the growing body. However, due to the challenges of isolating single postnatal/adult myocytes, how individual newborn cardiomyocytes acquire multiple aspects of the mature phenotype remains poorly understood. Here we implement large-particle sorting and analyze single myocytes from neonatal to adult hearts. Early myocytes exhibit wide-ranging transcriptomic and size heterogeneity that is maintained until adulthood with a continuous transcriptomic shift. Gene regulatory network analysis followed by mosaic gene deletion reveals that peroxisome proliferator-activated receptor coactivator-1 signaling, which is active in vivo but inactive in pluripotent stem cell-derived cardiomyocytes, mediates the shift. This signaling simultaneously regulates key aspects of cardiomyocyte maturation through previously unrecognized proteins, including YAP1 and SF3B2. Our study provides a single-cell roadmap of heterogeneous transitions coupled to cellular features and identifies a multifaceted regulator controlling cardiomyocyte maturation.

[1] Division of Cardiology, Department of Medicine, Johns Hopkins University School of Medicine, Baltimore, MD, USA. [2] Department of Biomedical engineering, Johns Hopkins University School of Medicine, Baltimore, MD, USA. [3] Institute for Cell Engineering, Johns Hopkins University School of Medicine, Baltimore, MD, USA. [4] Sanford Burnham Prebys Medical Discovery Institute, La Jolla, CA, USA. [5] Rutgers Institute for Translational Medicine and Science, New Brunswick, NJ, USA. [6] Division of Regenerative Medicine, Center for Molecular Medicine, Jichi Medical University, Tochigi, Japan. ✉email: ckwon13@jhmi.edu

Decades of advances in cellular and developmental cardiology have provided fundamental insights into understanding myocardial lineage specification in vivo, and this knowledge has been instrumental for producing cardiomyocytes (CMs) from pluripotent stem cells (PSCs)[1–3]. However, while newborn CMs continue to increase their volume and contractility through extensive morphological, functional, and metabolic changes until adulthood, PSC-derived CMs (PSC-CMs) are mired in an immature state even after long-term culture[4–6]. The lack of maturity significantly limits the scientific and therapeutic applications of PSC-CMs. Furthermore, despite a number of genes involved in CM maturation are associated with cardiomyopathies, little is known about its relevance to the initiation and progression of cardiac pathogenesis[6,7]. Thus, there is a significant need to understand biological processes underlying CM maturation in vivo.

CM maturation is a complex process essential for the heart to circulate blood to the rapidly growing body[8]. After terminal differentiation, CMs undergo binucleation/polyploidization around the first week of birth in mice. They gradually increase in size and become rectangular with uniformly patterned sarcomeres[9]. To efficiently propagate electrical activity, the plasma membrane invaginates into the cells and forms transverse tubules, enabling excitation-contraction coupling[10]. The myocytes become tightly connected via intercalated discs to allow simultaneous contraction. These events are accompanied with functional and metabolic changes including mature calcium handling, increased contractile force, and mitochondrial maturation and oxidative phosphorylation[8]. These multi-adaptive changes occur in the early postnatal period and continue until the adolescent/adult stages. However, it remains an open question whether these distinct processes occur in a coordinated fashion. Factors and pathways mediating these individual processes are poorly understood as well.

Previous large-scale meta-analyses provided a transcriptomic atlas of cardiac maturation, allowing us to determine gene regulatory networks and pathways involved in cardiac maturation[6]. The scope was, however, largely focused on prenatal stages, leaving the postnatal transcriptome dynamics unclear. Moreover, cell-to-cell variations—poorly understood in the field—could not be determined with bulk analysis. This issue can be addressed by single-cell RNA-sequencing (scRNA-seq) that enables comprehensive analysis of developmental and cellular trajectory and heterogeneity[11]. However, scRNA-seq is rarely utilized in myocyte biology due to technical difficulties associated with single-cell isolation of healthy, mature CMs.

We have recently demonstrated that large-particle fluorescence-activated cell sorting (LP–FACS) enables high-quality scRNA-seq and functional analysis of mature adult CMs[12]. Based on this, here we present high-quality scRNA-seq analysis of CMs isolated from neonatal to adult hearts. We demonstrate that newborn CMs are highly heterogeneous and progressively change the expression of genes regulating cellular hypertrophy, contractility, and metabolism until adulthood. By combining gene regulatory network analysis with mosaic gene deletion and PSC-CM assays at the single-cell level, we identify peroxisome proliferator-activated receptor (PPAR) coactivator-1 (PGC1) signaling as a multi-faceted regulator coordinating CM maturation via its targets Yap1 and SF3B2.

## Results

### CMs exhibit high levels of transcriptomic heterogeneity during postnatal maturation
Despite serving as a powerful tool in biology and medicine, the large size and fragility of mature CMs have limited the use of scRNA-seq in studying CM growth and disease[13]. Conventional sorting or microfluidic platforms result in damaged/ruptured myocytes due to inappropriate nozzle size/flow rate, leading to abnormal transcript reads and cell death. To address this, we recently tested LP–FACS and found that it allows isolation of healthy, mature myocytes, enabling both high-quality scRNA-seq and functional analysis[12]. Utilizing LP–FACS, we asked how postnatal CMs become mature cells at the single-cell transcriptome level. We harvested hearts from postnatal day (p) 0, p7, p14, p21, and p28 mice and dissociated CMs using standard Langendorff perfusion, followed by isolation of viable single myocytes with LP–FACS[12] (Fig. 1a). Curiously, UMAP based clustering[14,15] of p0–p28 single-cell samples show partial segregation (Fig. 1b), suggesting that significant numbers of cells at each stage may have similar transcriptome profiles as those of cells present at the other stages. We then used Monocle 2[16] to organize transcriptome profiles of CMs at different developmental stages based on transcriptomic similarities. The Monocle-based analysis produced a single pseudotime trajectory showing a progressive, unidirectional pattern of CMs over the course of maturation (Fig. 1c). Strikingly, individual cells from the same stages were found distributed broadly over the trajectory (Fig. 1d). When comparing the maturation score (as indicated by pseudotime) among different timepoints, we see that while average maturation scores increase with age, transcriptomic heterogeneity is maintained even at p28 myocytes (Fig. 1d). In agreement with this, similar levels of cell size heterogeneity were found in postnatal CMs (Fig. 1e).

We next examined the expression profiles of maturation-associated genes by plotting them along pseudotime, grouped by function. Expression levels of structural genes known to be upregulated in mature CMs, including *Myh6, Tnni3, Myom2*[8], were gradually increased (Fig. 1f). Calcium handling genes and ion channels including *Ryr2, Atp2a2, Casq1*, critical for contractility development, were continually upregulated while genes involved in the cell cycle, including *Cdk1, Cdk4, Ccnd1*[8,17], were downregulated (Fig. 1f). Conserved genes governing cellular hypertrophy, including *mTOR, Yap1, Igf1r*[18,19], were modestly downregulated or maintained in expression as they are known to regulate cell volume (Fig. 1f). Gene ontology (GO) analysis indicated that genes related to muscle contraction and cellular metabolism are highly regulated during the process (Fig. S1a). Consistently, hierarchical clustering showed that mitochondrial gene expression is gradually increased over time (Fig. S1b). Our single-cell approach quantitatively shows that postnatal CM maturation takes place in a continuous, but highly heterogeneous fashion.

### Gene network analysis predicts PGC1/PPAR signaling as a key regulator of cardiac maturation
The regulatory mechanisms underlying CM maturation are largely unknown. To gain mechanistic insights into upstream regulators governing postnatal CM maturation, we used Ingenuity Pathway Analysis (IPA) that infers regulators of differentially expressed genes by the knowledge base of expected effects between transcriptional regulators and their target genes[20]. Using the scRNA-seq dataset, we first generated gene regulatory networks with predicted upstream regulators by *p*-value and activation scores. We found that transcriptional cofactors and a group of nuclear receptors (PPARs, thyroid hormone receptors, retinoid receptors, etc.) are predicted to be upstream transcriptional regulators significantly affecting the overall gene expression changes during maturation (Fig. 2a, Fig. S1c–e). We further used a cut off of 10e-5 for the false discovery rate and ranked them by p-value. This analysis inferred the transcriptional cofactors PGC1α/β—two PGC1 isoforms—as among the most influential factors (Fig. 2b). While NFκBIα was predicted as the top regulator, its levels decreased

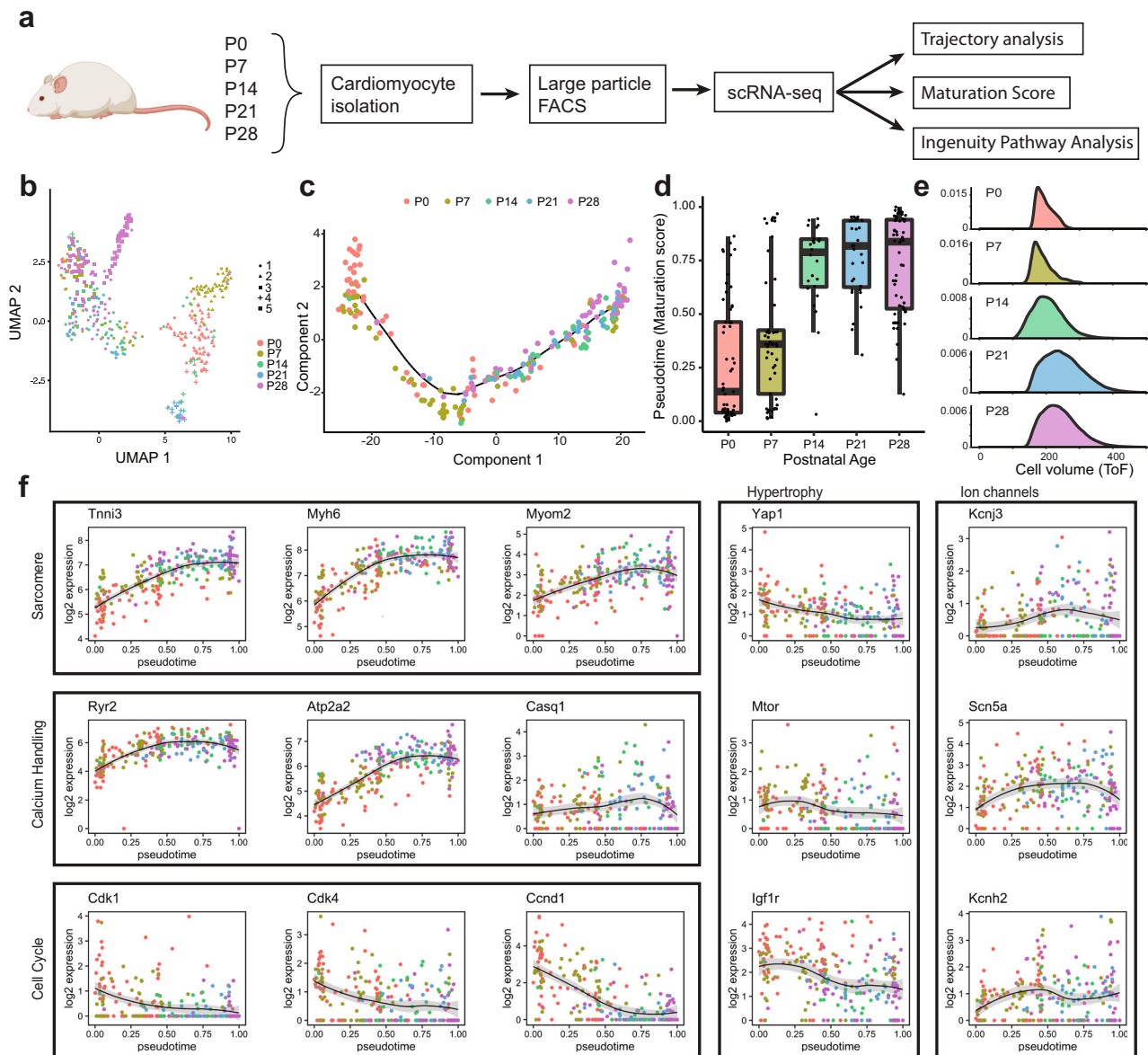

**Fig. 1 Postnatal CMs exhibit high levels of transcriptomic heterogeneity. a** Experimental design for scRNA-seq and computational analysis of CMs isolated from p0–p28 hearts. 300 CMs were sequenced after quality cutoff. **b** UMAP plot representation of p0 (red), p7 (gold), p14 (green), P21 (blue), p28 (purple) CMs. **c** Monocle-based developmental trajectory of p0–28 CMs. **d** Distribution of normalized pseudotimes (maturation scores) by age. **e** LP–FACS-based cell size analysis with time of flight (ToF). ToF tracts the time it takes the cell to get from one measured point of the flow cell to another. **f** Log expression of CM genes associated with structural maturation, calcium handling, cell cycle, hypertrophy, and ion channels plotted over pseudotime. Error bands represent a 95% Confidence interval. Box-and-whiskers plot represents the maxima, 75th percentile, median, 25th percentile, and minima.

over time (Fig. S1f). Thus, we focused on PGC1α/β and nuclear receptors for further analysis. Expression of the cofactors and nuclear receptors was gradually increased from embryonic to adult stages in vivo, with more pronounced upregulation after birth (Fig. 2c, d). An incremental expression pattern was also observed in most of the genes in long-term cultured PSC-CMs, but *PGC1/PPAR*α levels remained constantly low (Fig. 2c). This indicates that *PGC1/PPAR*α are misregulated in PSC-CMs and may be responsible for their maturation arrest[4,6].

**PGC1 is required cell-autonomously for postnatal CM growth and contractility development.** PGC1α/β are conserved transcriptional coactivators for PPARs and other nuclear receptors and known as central regulators of energy metabolism[21]. The two isoforms show extensive sequence homology with functional

redundancy[22,23]. In fact, embryonic deletion of either PGC1α or PGC1β does not affect heart formation, but the double knockout results in lethality soon after birth with small hearts accompanied by mitochondrial defects[22]. However, these studies deleted the alleles globally or in embryonic stages, leaving their cell-autonomous, postnatal role unknown. Based on the single-cell bioinformatics prediction and low levels of expression in PSC-CMs, we hypothesized that PGC1 intrinsically mediates postnatal maturation of CMs. To test this, we generated conditional mosaic knockout (cmKO) mice, which avoids non-cell-autonomous effects and early lethality caused by global or conditional deletion (Fig. 3a). For this, we generated *PGC1α/β^flox/flox; Ai9* mice and administered AAV vectors expressing Cre specifically in CMs (AAV9-cTnT-iCre) at p0. In this system, cmKO cells are generated in neonatal CMs and identified by RFP expression. We titrated AAV vector particles and injected subcutaneously a dose

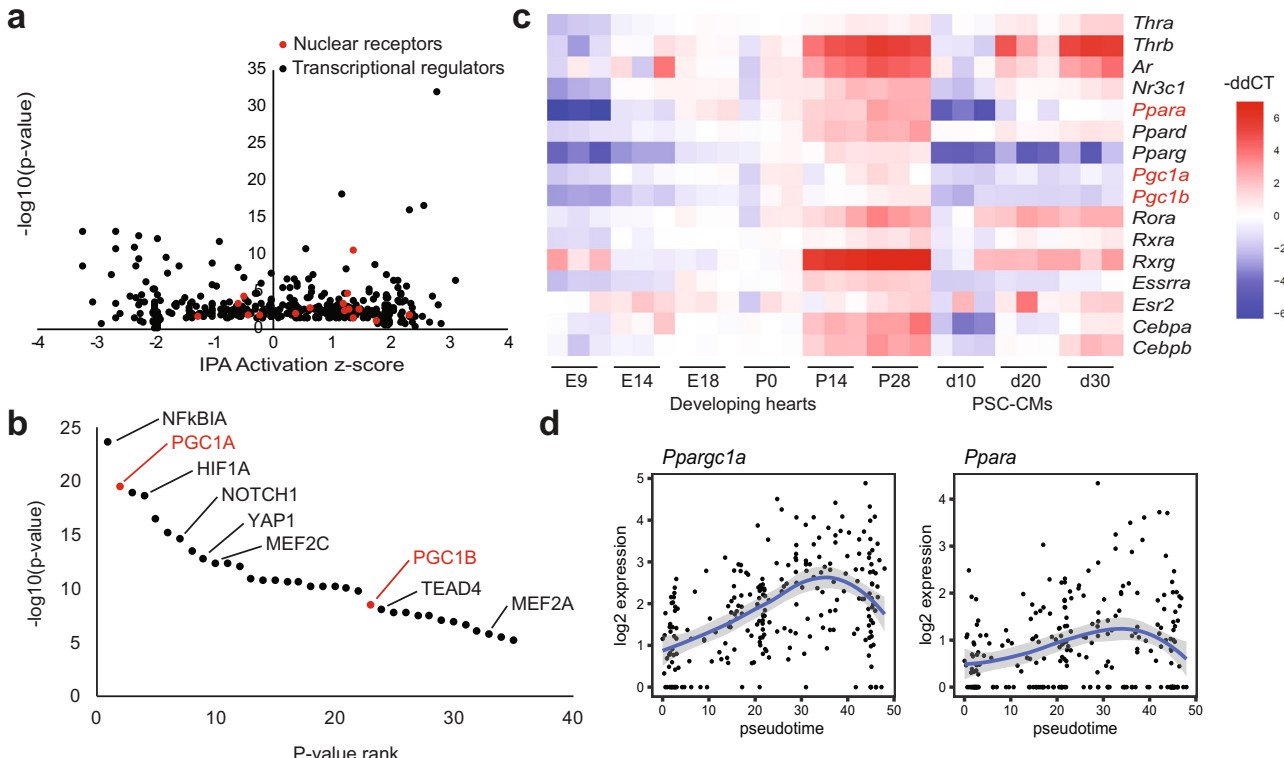

**Fig. 2 PGC1/PPAR is a predicted key upstream regulator of CM maturation. a** Top transcriptional regulators plotted by p-value and IPA activation z-score with nuclear receptors highlighted in red. **b** P-value ranking of top upstream regulators of CM maturation with two PGC1 isoforms highlighted in red. **c** Heatmap of gene expression of PGC1 and nuclear receptors in developing mouse hearts and cultured PSC-CMs, quantified by qPCR. **d** Expression trends over pseudotime of PGC1 and PPARα in postnatal CMs. Error bands represent a 95% confidence interval.

of 2e10 genome copies per mouse that results in a mosaic heart with 5–10% RFP$^+$ myocytes. The resulting RFP$^+$ myocytes showed efficient deletion of $PGC1\alpha/\beta$, quantified by qPCR (Fig. S2a–e). We made transverse sections and analyzed RFP$^+$ cells with α-actinin staining. RFP$^+$ cells appeared smaller than neighboring myocytes in size (Fig. 3b). To precisely quantify size, we dissociated CMs using Langendorff perfusion and measured the areas of RFP$^+$ and RFP$^-$ CMs after plating at low density (Fig. 3b). RFP$^-$ cells showed a heterogeneous but progressive increase in size over time, consistent with single-cell LP–FACS cell volume analysis (Fig. 3c, cyan columns). While RFP expression itself did not affect cell size (Fig. S2f), we observed that RFP$^+$ cells remain persistently smaller compared to RFP$^-$ cells. (Fig. 3c, red columns). This suggests the requirement of PGC1 in cellular hypertrophy. Next, we tested whether PGC1 deficiency affects intact CM contractile function by video microscopy. RFP$^+$ cells showed significantly lower fractional shortening and contraction velocity (Fig. 3d–f, Fig. S2g–j). Their contractile properties were further assessed by measuring calcium transients with the ratiometric dye Fura-2 AM. Consistent with the sarcomere shortening data, RFP$^+$ cells showed significantly lower peak Ca$^{2+}$ amplitude and higher time to baseline than RFP$^-$ cells (Fig. 3g–i, Fig. S2k–n). These kinetics suggest that PGC1-deficient myocytes develop less mature calcium cycling apparatus for Ca$^{2+}$ release and re-sequestration. However, t-tubule structures remained intact in PGC1-deficient CMs (Fig. S2o). Together, our mosaic gene deletion approach reveals a cell-autonomous, required role of PGC1 in cellular hypertrophy and contractility development of postnatal CMs at the single-cell level.

**PGC1 regulates genes affecting cell size, calcium handling, and mitochondrial activity.** Given the crucial role of PGC1 in

developing myocyte hypertrophy and contractility, we investigated how PGC1 mediates these processes at the single-cell transcriptome level. After deleting PGC1 as above, we isolated RFP$^+$ (PGC1 cmKO) cells by LP-FACS from p7–p28 hearts and sequenced 768 cells (300 RFP$^-$ CMs and 326 RFP$^+$ CMs) passing quality control with a mean of 69,868 UMIs per cell (allotted 1 million reads per cell). The resulting trajectory reconstructed by Monocle with control and cmKO cells showed that PGC1-deficient CMs become less heterogeneous at p7 (Fig. 4a, b). They also maintained lower maturation scores throughout the stages (Fig. 4b), which is consistent with the failure to increase in contractility and size.

To determine how PGC1 signaling mediates these processes, we used fuzzy clustering to group the temporal trends of individual gene expression in both control and PGC1 cmKO CMs (Fig. 4c, d). We initially selected upregulated and downregulated clusters for each group and subsequently performed overlap analysis. Notably, the analysis showed that <7.6% or 16.2% of genes overlap between control and PGC1 cmKO CMs in upregulated or downregulated clusters, respectively (Fig. 4c, d). This suggests that gene regulatory networks associated with maturation have been severely disrupted in PGC1-deficient CMs. GO analysis of differentially expressed genes (329 upregulated, 255 downregulated) showed that muscle fiber/cell development and mitochondrial/electron transport chain processes are significantly impaired in PGC1 cmKO cells (Fig. 4e).

**PGC1/PPARα activation increases size, contractility, and mitochondrial activity of PSC-CMs.** Since PGC1 is required for CM hypertrophy and contractility, and its levels and activity remain low in PSC-CMs[6], we further investigated if increased levels of PGC1 can promote the maturation of PSC-CMs. For

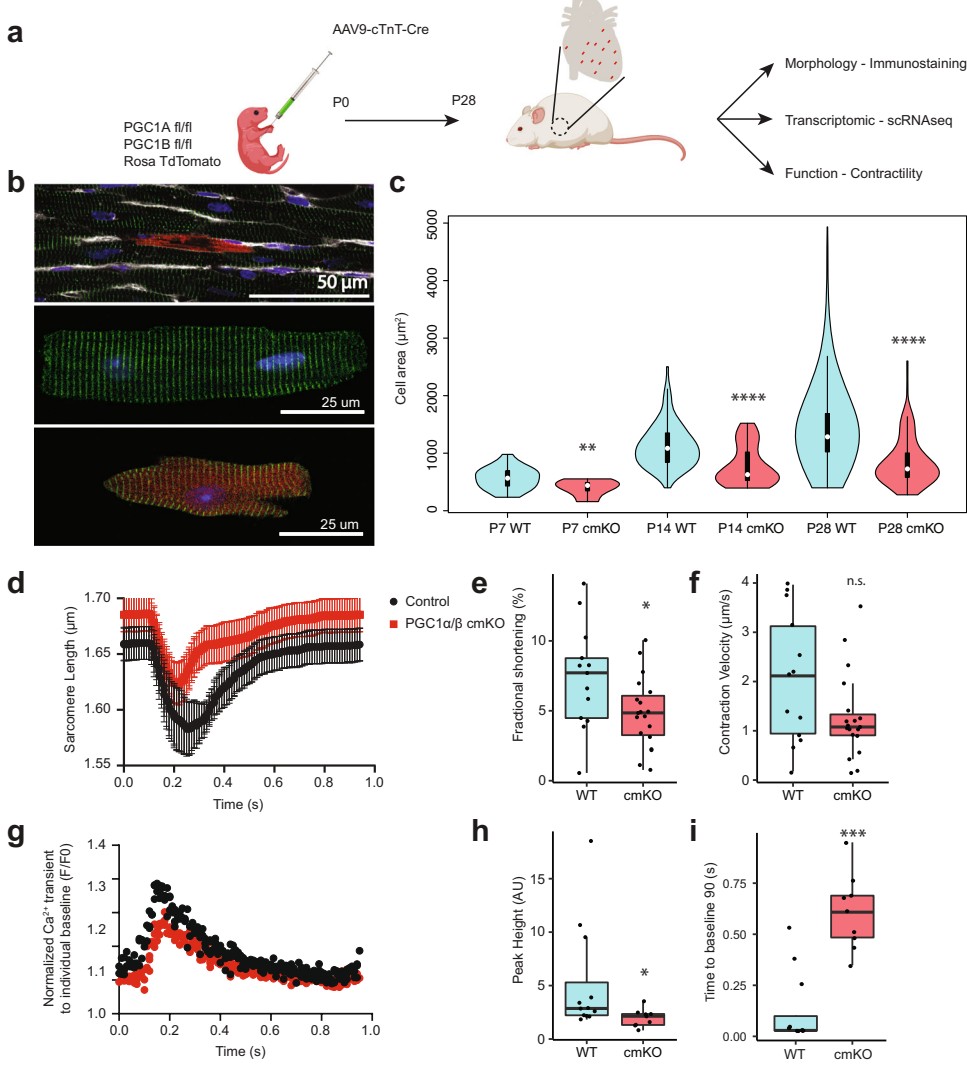

**Fig. 3 PGC1 is required for CM hypertrophy and contractility development. a** Experimental scheme showing generation and analyses of a cmKO heart achieved by injection of AAV9-cTnT-Cre into *PGC1α/β^flox/flox*; *Ai9* mice at p0. **b** Heart slice showing a cmKO myocyte in the myocardium (top) and dissociated control (middle) and cmKO (bottom) myocytes. **c** Violin plots of cell area distributions in control (blue) and cmKO (red) CMs at p7 (*p*-value = 0.003), p14 (*p*-value = 2.5e−7), p28 (*p*-value = 2.2e−16). *n* = 44, 11, 90, 49, 522, 132 (left to right). ANOVA followed by post hoc Bonferroni tests were used. **d**–**f** Sarcomere shortening data with the average trace, fractional shortening (*p*-value = 0.047) and contraction velocity (*p*-value = 0.07). Control *n* = 13, cmKO *n* = 19. The student's *t*-test was used. **g**–**i** Calcium handling with average calcium trace, peak height (*p*-value = 0.023, AU = arbitrary units), and time to baseline 90 (*p*-value = 0.0006). Control *n* = 13, cmKO *n* = 9. Mann–Whitney–Wilcoxon test and the Student's *t*-test were used. Box-and-whiskers plot represents the maxima, 75^th percentile, median, 25th percentile, and minima. *P*-value: *<0.05,**<0.01,***<0.001,****<0.0001.

this, we first differentiated PSCs into CMs using established protocols[6,24] (Fig. 5a) and increased PGC1 levels by expressing GFP-PGC1[25]. We found that GFP^+ CMs became significantly larger than GFP^− CMs after transfection (Fig. S3a). The hypertrophic growth was recapitulated by pharmacological stimulation with pyrroloquinoline quinone (PQQ), a small molecule that is known to activate PGC1 signaling[26–28] (Fig. 5b, c, Fig. S3c). PGC1 regulates various cellular processes via binding to the transcription factors PPARs[21,23,29]. Given that PPARα remains downregulated in PSC-CMs (Fig. 2c) and is known to have important roles in myofibril structure and contractility[30,31], in line with the expression and role of PGC1, we tested if increased PPARα signaling can affect PSC-CM maturation. To do this, we stimulated PSC-CMs with the PPARα-specific ligand WY14643[32]. Similar to activating PGC1 signaling, WY14643-treated PSC-CMs significantly increased cell size (Fig. 5b). Individual treatment significantly improved contractile force as well,

measured by traction force microscopy, which was further increased when combined (Fig. 5d, Fig. S3b). This phenotype was accompanied with a significant increase in mitochondrial density and respiration and a decrease in glycolysis, determined by electron microscopy and Seahorse assay (Fig. S3d–h). Together, these data suggest that PGC1/PPARα have an instructive role in cellular hypertrophy and contractility development in PSC-CMs.

**PGC1/PPARα signaling promotes CM hypertrophy and contractility development via YAP1 and SF3B2, respectively.** To determine how PGC1/PPARα mediates CM growth, we analyzed expression levels of conserved cell size regulators *mTOR, Yap1, Igf1*. Among these candidates, we found that *Yap1*, a transcriptional effector for Hippo signaling with crucial roles in cardiac regeneration[33], is markedly downregulated in PGC1 cmKO cells (Table S1). To test if Yap1 affects CM size in vivo, we generated Yap1 cmKO cells in postnatal hearts as shown in Fig. 3a, and

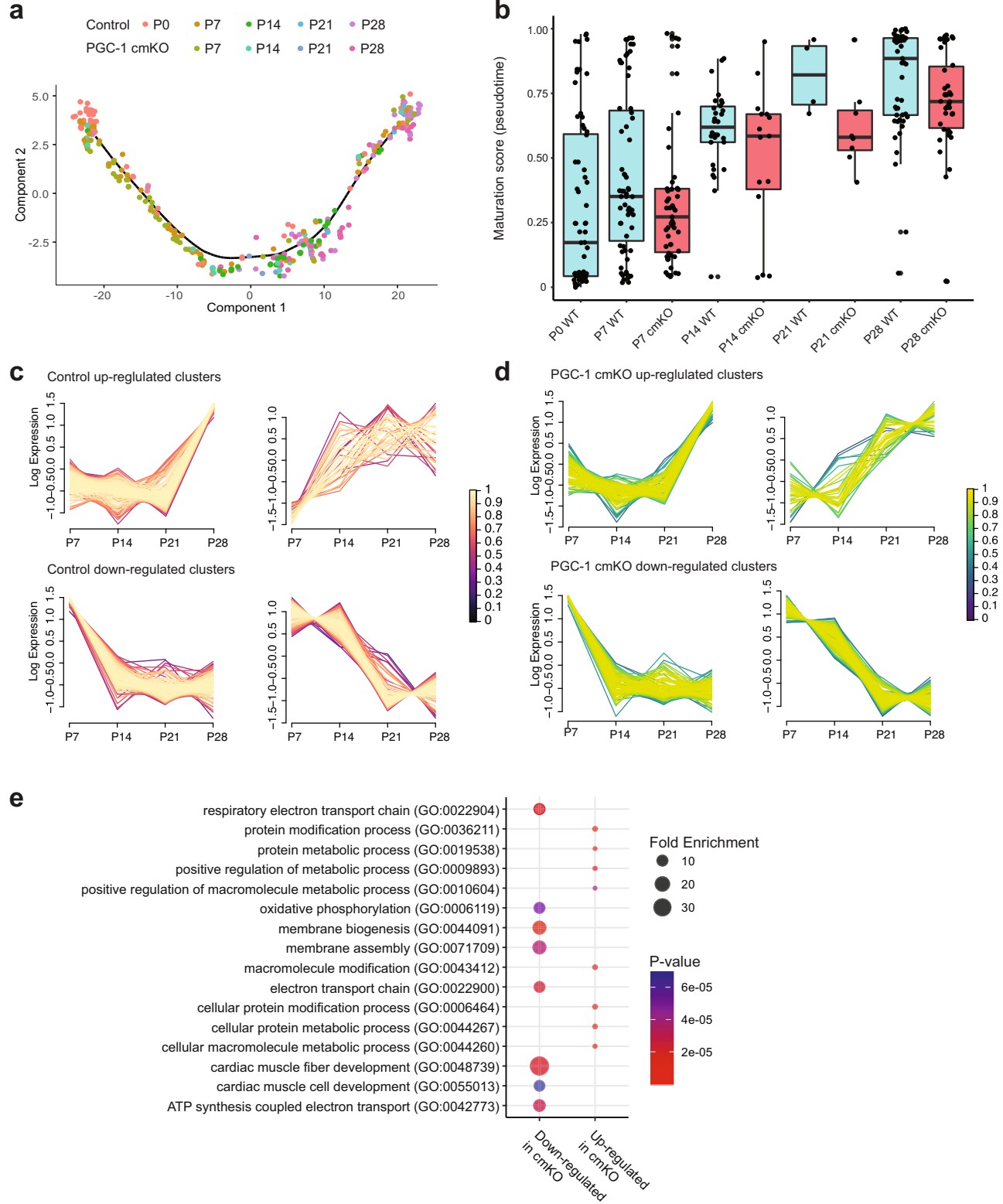

**Fig. 4 PGC1-deficiency leads to maturation defects in postnatal CMs and affects gene regulatory networks controlling muscle development and mitochondrial process. a** Single-cell transcriptomic trajectory of control and PGC1 cmKO CMs. 300 control and 328 cmKO CMs were used after quality cutoff. **b** Distribution of pseudotime maturation scores analyzed from p0-p28. **c** Fuzzy clustering with selected clusters for upregulated (top) and downregulated (bottom) genes. Color indicates the membership score of each gene in the cluster. **d** Fuzzy clustering of PGC1 cmKO CMs showing upregulated (top) and downregulated (bottom) clusters. **e** GO term visualization by fold enrichment (dot size) and *p*-value (dot color). Box-and-whiskers plot represents the maxima, 75th percentile, median, 25th percentile, and minima.

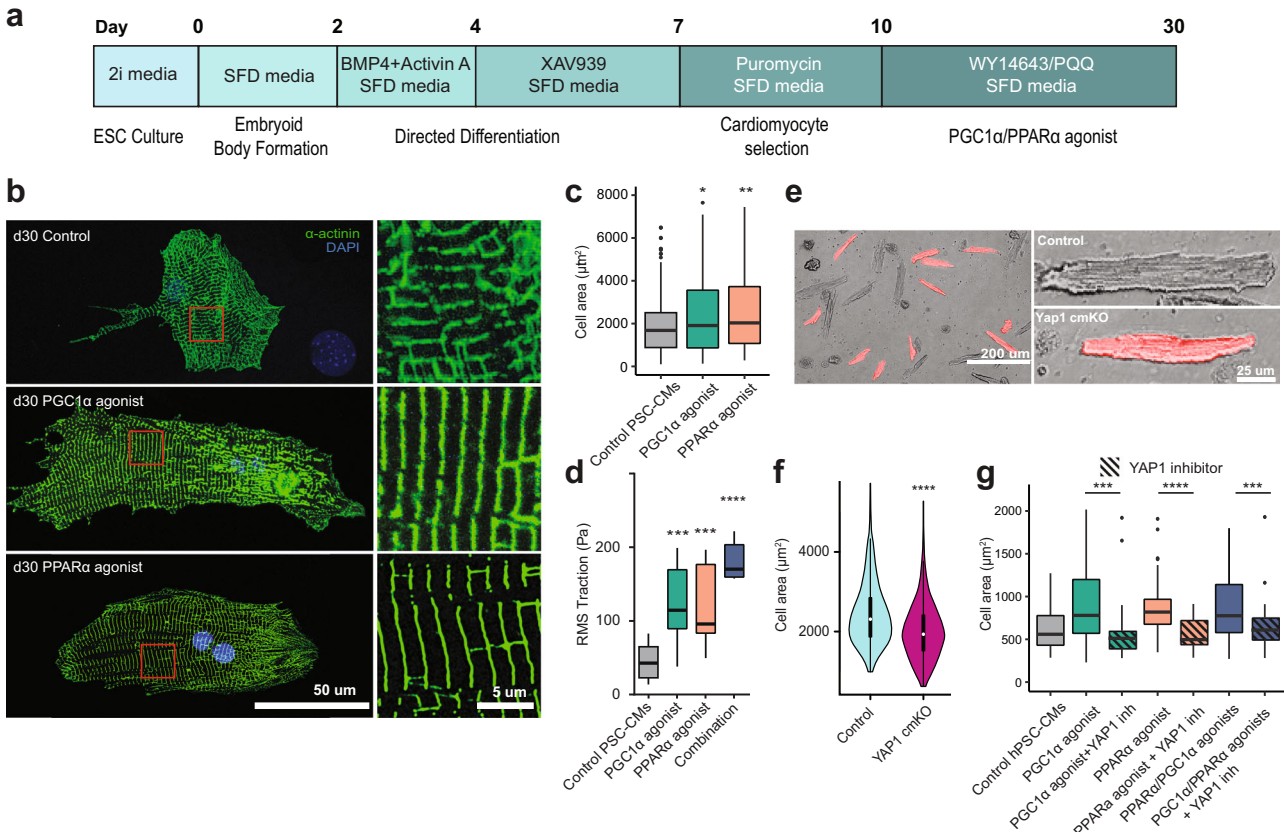

**Fig. 5 PGC1/PPARα promotes PSC-CM hypertrophy through YAP1. a** PSC-CM differentiation protocol and treatment timeline showing PGC1α/PPARα treatment following differentiation until d30. **b** Control and PGC1α/PPARα agonist-treated mouse ESC-CMs at day 30 (dissociated and replated at low density). The CMs were stained with α-actinin antibody (green) and dapi (blue). Insets show magnified views of boxed areas shown in the left. PQQ 10 μM, WY14643 1 μM. **c** Quantification of cell area of mouse ESC-CMs treated with PGC1α (green)/PPARα (peach) agonists for 15 days. Control $n = 192$, PGC1 agonist $n = 162$ (p-value = 0.012), PPARα agonist $n = 118$ (p-value = 0.0015). ANOVA followed by post hoc Bonferroni test was used. **d** Fourier transforms traction microscopy showing RMS traction in pascals of control and treated mouse ESC-CMs indicates increased contractility and force production in agonist-treated PSC-CMs. $n = 6$. **e** Representative images of control (RFP⁻) and Yap1 cmKO (RFP⁺) CMs isolated from p32 hearts. **f** Quantification of cell area of control and Yap1 cmKO CMs. Control (blue) $n = 421$, YAP1 (magenta) cmKO $n = 335$. (p-value = 1.e−14) The student's t-test was used. **g** Cell area measurements for human ESC-CMs treated with PGC1α/PPARα agonists in combination with YAP1 inhibitor ((R)-PFI-2 1 μm). $n = 25, 26, 41, 52, 40, 31, 22$ by column (p-values = 0.00092, 1.7e−6, 0.017). Box-and-whiskers plot represents the maxima, 75th percentile, median, 25th percentile, and minima.

analyzed their growth. Notably, the cmKO CMs became significantly smaller than normal CMs (Fig. 5e, f), indicating that Yap1 may mediate PGC1/PPARα signals for CM hypertrophy. To test this, we chemically blocked YAP1 transcriptional activity with the Yap1 inhibitor (R)-PFI-2[34] in PSC-CMs stimulated with PGC1/PPARα activators. We found that blocking YAP1 activity indeed abolished the increase in cell size induced by PGC1/PPARα activation (Fig. 5g). These data suggest that YAP1 is required for PGC1/PPARα to promote CM hypertrophy. A previous chromatin immunoprecipitation (ChIP) study reported that PGC1 binds to the *YAP1* locus in hepatic cells[35]. Similarly, we found that PGC1/PPARα are physically associated with the promoter region of *Yap1* in postnatal CMs, determined by ChIP with PGC1/PPARα antibody followed by qPCR (ChIP-qPCR) (Fig. S4). This indicates that *Yap1* may be directly regulated by PGC1/PPARα.

Since PGC1/PPARα signals promoted CM contractility, we next sought to identify genes mediating the process. To do this, we selected 148 genes significantly downregulated in p7 PGC1 cmKO CMs (Table S2) and performed a single-cell high-throughput functional assay with PSC-CMs[36–38] treated with PPARα-specific ligands (Fig. 6a). Mitochondrial and predicted genes were removed for this assay. Single-cell analysis of calcium

handling revealed that ligand-treated PSC-CMs have significantly shorter (~30 ms) calcium transient duration (CTD) as compared to vehicle-treated cells (DMSO) (Fig. 6b, c). Notably, calcium transient peak rise time was shorter, and CTD50 and 75 were decreased (Fig. 6d–f), thereby suggesting that calcium handling properties are enhanced in ligand-treated cells. Next, to identify downstream effectors mediating the CTD shortening, we stimulated PSC-CMs with PPARα ligands and applied a library of siRNAs (4 siRNAs/gene) targeting genes regulated by PGC1 signaling (Fig. 7a). We found that the ability of PPARα signaling to shorten CTD is significantly impaired when targeting genes encoding *SF3B2/SAP18* (RNA splicing factors)[39,40], and *TIMM50* (a mitochondrial translocase)[41] (Fig. 7b–c, Fig. S5b). In particular, *SF3B2* knockdown showed the most significant effect on CTD (Fig. 7d–i). Consistent with the impaired calcium handling, PSC-CMs deficient of the identified factors exhibited reduced contractility, with SF3B2 being most severely affected (Fig. 7j, Fig. S5a). These data suggest that SF3B2 is a key mediator of PGC1/PPARα signaling for the functional maturation of PSC-CMs. Our ChIP-qPCR analysis further showed that PGC1/PPARα physically interacts with the *Sf3b2* promoter (Fig. S4c, d), implying that *Sf3b2* may be directly regulated by PGC1/PPARα as well.

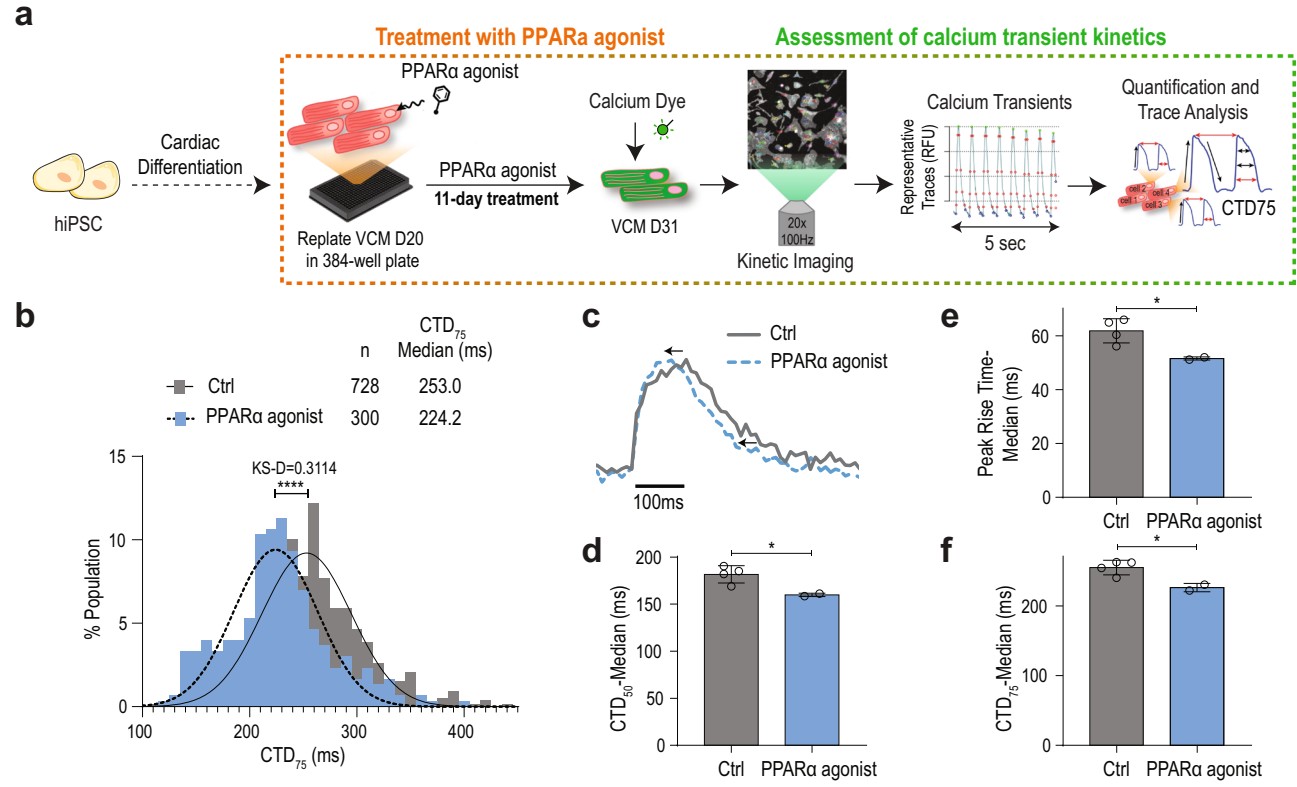

**Fig. 6 PGC1/PPAR activation improves calcium handling in PSC-CMs. a** Experimental diagram of human PSC-CM differentiation, agonist treatment, and calcium function analyses. **b** Distribution of calcium transient duration (CTD) 75 of single ESC-CMs shows a shorter CTD75. **c** Sample calcium transient trace for control (gray) and PPARα agonist (dashed blue) groups. **d–f** median peak rise ($p$-value = 0.039), $CTD_{50}$ ($p$-value = 0.034), and $CTD_{75}$ ($p$-value = 0.026) times. control $n = 728$, PPARα agonist $n = 300$ cells, $p$-values: *<0.05, ****<0.001. The student's $t$-test was used. Error bars represent the standard deviation.

## Discussion

In the present study, we investigated how individual CMs give rise to mature cells, a fundamental, yet poorly understood event. We have demonstrated that CMs are heterogeneous in transcriptome and size during postnatal stages, when significant levels of adaptive changes occur, with variable expression levels of genes responsible for cell growth and contractility. We found that PGC1 signaling, whose activity is increased until adulthood in vivo but remained low in PSC-CMs, is necessary to regulate genes crucial for hypertrophy and contractility in addition to mitochondrial genes, albeit in a heterogeneous manner among cell populations. These findings provide fundamental and mechanistic insights into the postnatal maturation of single CMs, coordinated by PGC1 signaling (Fig. S5c).

Earlier transcriptome analysis of developing hearts suggested that CMs develop unidirectionally towards a more mature state through discrete developmental stages[6], yet it remains unknown if this represents homogeneous maturation of individual myocytes. It is intriguing that CMs exhibit and maintain a high level of developmental heterogeneity throughout postnatal maturation stages. Indeed, a small subset of early myocytes showed mature transcriptomes as early as P7, whereas some late myocytes still have transcriptomes resembling immature cells. This suggests that postnatal CMs mature at different rates, and achieving full maturation of individual myocytes may not precisely follow the developmental timeline or may not occur in all myocytes. This is supported by heterogeneous cell size and the presence of myocytes expressing *Myh7* and *Tnni1* at P28, considered not to be expressed in mature myocytes. It would be important to further investigate where the immature or mature cells are located and if

the immature cells represent a small subset of proliferative myocytes present in adult hearts.

Understanding the factors and mechanisms underlying cardiac maturation is of great importance, but there is very little information available at this point. This could be attributed to the difficulty of staging and defining milestones of the process that takes place over a long period of time. In fact, a recent study demonstrated the importance of stage-specific gene regulation in CM maturation[42]. Our study suggests that *PGC1* is upregulated after birth and its activity is required and sufficient to promote CM maturation. Intriguingly, while PGC1 signaling is known to play a critical role in controlling mitochondrial biogenesis and cellular metabolism[21], our single-cell analysis showed that it regulates multiple aspects of cellular events, including cell size, calcium handling, and contractility in addition to oxidative phosphorylation, during cardiac maturation. This is particularly surprising given its conserved and universal role in energy metabolism. These findings suggest that PGC1 signaling functions as a multifaceted regulator in postnatal CM maturation. Interestingly, *t*-tubules were detected in PGC1 mutant myocytes, suggesting that the morphogenetic event is not adversely affected by the lack of PGC1 signaling. However, a recent paper demonstrated the necessity of a 3D environment in generating a *t*-tubule-like structure in PSC-CMs[43], suggesting that *t*-tubule formation could be an adaptive process in response to microenvironmental changes.

A number of previously unrecognized genes were found regulated by PGC1/PPARα in postnatal stages. In particular, splicing factors SF3B2/SAP18 were required for the functional maturation of PSC-CMs. RNA splicing is an important post-transcriptional

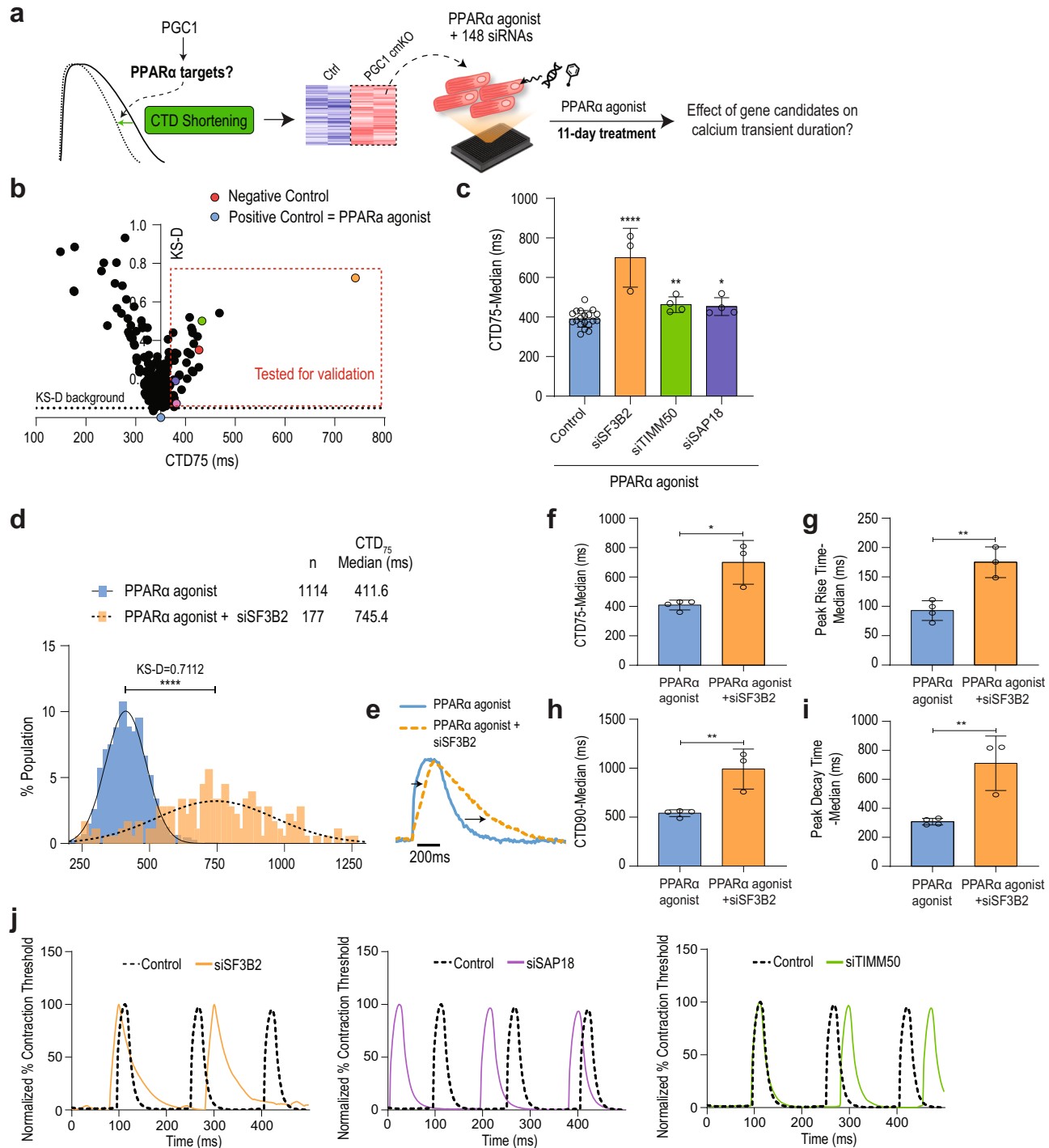

**Fig. 7 High-throughput single-cell functional assay shows PGC1/PPAR regulates calcium handling and contraction via SF3B2 in PSC-CMs. a** Experimental design for siRNA-based functional assay with 148 genes downregulated in PGC1-deficient CMs. **b** siRNA assay results with Kolmogorov-Smirnov distance (KS-D) and CTD75. Untreated or PPARα agonist-treated hPSC-CMs were used as negative or positive control, respectively. **c** Median CTD75 for validated hit siRNAs. Control $n = 16$ (siSF3B2 (orange) $p$-value $> 0.0001$ $n = 3$, siTIMM50 (green) $p$-value $= 0.005$ $n = 4$, siSAP18 (purple) $p$-value $= 0.0145$ $n = 4$) **d** Distribution of CTD75 in PPARα agonist-treated human PSC-CMs with(blue)/without(orange) siRNA targeting *SF3B2*. **e** Representative traces of hPSC-CM calcium transient for PPARα agonist-treated and PPARα agonist with siSF3B2. **f–i** Calcium transient parameters showing functional effects of *SF3B2* knockdown. **j** Traces showing effects of validated hit siRNAs on contractility by normalized percentage of contraction threshold. $p$-values: *$<0.05$,**$<0.1$,***$<0.01$,****$<0.001$. ANOVA with post hoc Bonferroni test was used for multiple comparisons. The student's $t$-test was used. Error bars represent standard deviation.

mechanism, and its abnormal regulation is closely associated with human diseases including heart disease[44,45]. While splicing factors were shown to promote neuronal maturation[46], their role in the context of CM maturation remains to be determined. We found that PGC1/PPARα regulates *Yap1*, a conserved transcriptional regulator of organ/cell size[18] during postnatal CM development. The role of Yap1 has been extensively studied on CM proliferation for heart repair[47], but growing evidence suggests its critical role in cellular hypertrophy[19,48]. Similarly, our data showed that *Yap1* is a key mediator of PGC1/PPARα for CM growth. This is consistent with the finding that YAP1 directly regulates cell volume via regulating cytoplasmic pressure, and this occurs independent of mTOR[18,19]. A previous study suggested that Yap1 is not required for physiological cardiac hypertrophy[49], but this discrepancy could be due to methodological or technical differences in cell size quantification. Further investigation would be necessary to precisely dissect the effect of Yap1 on CM hypertrophy.

It is also worth pointing out that, while we used PQQ to pharmacologically activate PGC1 signaling, PQQ, as a redox cofactor, may also have other biological effects on mammalian cells that have not been fully characterized.

PGC1/PPAR signaling regulates numerous genes involved in mitochondria biogenesis and cellular metabolism. Curiously, fatty acid treatment was shown to enhance structural and functional maturation of PSC-CMs[50], similar to PGC1/PPARα activation. However, GO analysis suggests that the underlying mechanisms may be different: while fatty acids regulate genes and pathways involved in fatty acid β-oxidation and lipid synthesis[50], PGC1 affects cardiac muscle development and oxidative phosphorylation (Fig. 4e). Nevertheless, these suggest the importance of cellular metabolism/energy production in PSC-CM maturation. Future work would be needed to understand if and how PGC1/PPARα regulates energy metabolism via YAP1/SF3B2.

PSC-CMs have great potential for a wide range of preclinical and clinical applications, including cardiac disease modeling, drug discovery, and regenerative medicine. The resulting myocytes, however, are prematurely arrested at late embryonic stage in culture[4,6], and the inability to produce mature myocytes from PSCs is a major hurdle for their broad applicability. We found that PGC1/PPAR signaling remains inactive even in long-term-cultured PSC-CMs, and its activation promotes their size, contractility, and metabolism, key features of myocyte maturation. This suggests that the developmental arrest is at least in part attributed to the lack of PGC1/PPAR activity in PSC-CMs. This is consistent with the structural and functional arrest observed in PGC1 KO CMs in vivo. Knowing the complexity of cardiac maturation, our finding is expected to help us further investigate gene regulatory networks and barriers controlling the distinct processes of cardiac maturation.

## Methods

**Animals and PSC culture**. *PGC1α/β flox*, *Ai9*, *Yap1flox* mice[22,51–53] were obtained from the Jackson Laboratory. All protocols involving animals followed U.S NIH guidelines and were approved by the animal and care use committee of the Johns Hopkins Medical Institutions. Animals were housed at room temperature in ventilated racks containing automatic watering. They were exposed to a cycle of 14 h of light then 10 h of dark. CMs were differentiated from mouse and human ESCs (E14 and H9, WiCell) as described with consent being received from donors[6,24]. Briefly, mouse ESCs were cultured in gelatin-coated flasks with stem cell maintenance media (GMEM + 10% FBS with 3 µm Chir99021, 1 um PD98059, glutamax, non-essential amino acids, and sodium pyruvate. To differentiate, embryoid bodies were formed by plating 80,000 cells/mL into uncoated dishes. The medium used for mouse PSC-CM differentiation and PSC-CM culture contained 75% IMDM, 25% Ham's F12 (Cellgro) with B27 without vitamin A, N2, BSA, glutamax, Penicillin/Streptomycin, L ascorbic acid, and a-monothioglycerol. 48 h later, embryoid bodies were collected and induced for 48 h with Bmp4 and Activin A, then dissociated and replated with XAV939 (Sigma).

**Generation of hiPSC-CMs**. Id1 overexpressing hiPSCs[36] were dissociated with 0.5 mM EDTA (ThermoFisher Scientific) in PBS without CaCl2 and MgCl2 (Corning) for 7 min at room temperature. hiPSC were resuspended in mTeSR-1 media (StemCell Technologies) supplemented with 2 µM Thiazovivin (StemCell Technologies) and plated in a Matrigel-coated 12-well plate at a density of 3 × 105 cells per well. After 24 h after passage, cells were fed daily with mTeSR-1 media (without Thiazovivin) for 3–5 days until they reached ≥90% confluence to begin differentiation. hiPSC-CMs were differentiated as previously described[54]. At day 0, cells were treated with 6 µM CHIR99021 (Selleck Chemicals) in S12 media[55] for 48 h. At day 2, cells were treated with 2 µM Wnt-C59 (Selleck Chemicals), an inhibitor of WNT pathway, in S12 media. 48 h later (at day 4), S12 media is fully changed. At day 5, cells were dissociated with TrypLE Express (Gibco) for 2 min and blocked with RPMI (Gibco) supplemented with 10% FBS (Omega Scientific). Cells were resuspended in S12 media supplemented with 4 mg/L Recombinant Human Insulin (Gibco) (S12 + media) and 2 µM Thiazovivin and plated onto a Matrigel-coated 12-well plate at a density of 9 × 10⁵ cells per well. S12 + media was changed at day 8 and replaced at day 10 with RPMI (Gibco) media supplemented with 213 µg/µL L-ascorbic acids (Sigma), 500 mg/L BSA-FV (Gibco), 0.5 mM L-carnitine (Sigma), and 8 g/L AlbuMAX Lipid-Rich BSA (Gibco)(CM media). Typically, hiPSC-CMs start to beat around day 9–10. On day 15, cells were purified with lactate media (RPMI without glucose, 213 µg/µL L-ascorbic acid, 500 mg/L BSA-FV, and 8 mM Sodium-DL-Lactate (Sigma), for 4 days. At day 19, media was replaced with CM media.

**Heart dissociation, sorting, and immunostaining**. Harvested hearts were placed in a Langendorff setup and perfused with a Type II Collagenase and Protease digestion buffer and stepped up with Calcium to 1 mM. Viable CMs were sorted using LP-FACS[12]. The Union Biometrica COPAS Flow Platform was used to select CMs based on extinction, time of flight, and red fluorescence. For staining, hearts were flash frozen into O.C.T Compound blocks (Fisher Healthcare 23-730-571) and stored at −80 C. Samples were sliced into 10 um sections using a cryo-microtome and placed on glass slides. Primary antibodies against 1:500 dilution α-actinin (Abcam ab227074) were used along with 1:1000 dilution DAPI and 1:200 dilution Wheat Germ Agglutinin counterstains. A Leica SP8 confocal microscope running Leica Application Suite X was used to image slides with a ×63 oil immersion objective.

**Calcium handling and shortening analysis**. Sorted cells were plated onto laminin-coated glass slides then loaded with Fura2AM Ca²⁺ dye. They were analyzed with the IonOptix imaging system and IonWizard software[56].

**Single-cell RNA sequencing**. For scRNA-seq, individual cells were fluorescently sorted into 96 well plates using the Union Biometrica COPAS Flow Platform, which were frozen on dry ice and stored at −80 C. We used SCRB-seq to prepare libraries[12,57]. We sequenced 300 control and 328 PGC1 cmKO CMs that passed our quality cutoff. Paired-end reads were sequenced using an Illumina NextSeq500 then mapped with STAR and featureCounts-based zUMIs[58]. Monocle 2 and Seurat v3.0 were used for trajectory, differential expression, and UMAP clustering.

**Chromatin Immunoprecipitation qPCR**. For ChIP, hearts were isolated, washed, and minced then fixed in 1% paraformaldehyde crosslinking buffer. Tissue chunks were homogenized, and DNA was fragmented using 20 cycles of 30 s on then 30 s off of 50% power sonication. Protein G magnetic beads were incubated with the sample for 1 h. Antibody (PGC1α Santa Cruz, PPARα Abcam ChIP-grade ab227074) was incubated at 4 C overnight. Protein G magnetic beads were incubated for 1 h then pulled down. Chromatin was eluted from the beads then uncrosslinked by incubating overnight at 65 °C. DNA was purified using phenol-chloroform following RNase and Proteinase K treatment. Primers (Supplementary Data 1) were designed using Primer3 targeting promoter regions from the transcriptional start site to 1 kb upstream using SYBR Primer master mix (Thermofisher).

**Mitochondrial functional assay**. Respiration rates were measured with Seahorse XFe96 Analyzer. CMs were plated at 1.5 × 10⁴ cells per well of a 96 well XF96 Cell Culture Microplate (Aligent Technologies) and cultured for 3 days. One hour before the assay, the medium was changed to RPMI without phenol red supplemented with sodium pyruvate. The Seahorse Extracellular Flux Assay Kit was used with the Mito Stress Test protocol. Inhibitor final concentrations were Oligomycin (2.5 µM), FCCP (1 µM), and Rotenone (2.5 µM) + Antimycin A (2.5µM).

**Electron microscopy**. D30 PSC-CMs were fixed by 2% GA in 0.075 cacodylate with 5 mM MgCl₂ overnight at 4 °C. Samples were rinsed three times for 15 min with a 3% sucrose buffer then treated with 2% osmium and 1.5% KFeCN₆ for 2 h at 4 °C. They were rinsed with 0.1 M maleate buffer pH 6.2 with 3% sucrose three times for 10 min then 2%UA in maleate/sucrose buffer for 1 h without light. Samples were dehydrated in an ethanol ladder stepping up from 30 to 100% at 5 min each. They were treated with propylene oxide then EPON resin with catalyst

overnight with rocking then EPON resin with a catalyst for 2 h then placed in a 60 C oven for 48 h. Sections were cut using a Diatome diamond knight collected on 2 × 1 mm formvar-coated slot grids and stained with uranyl acetate followed by lead citrate. A Hitachi H-7600 TEM operating at 80 kV. An AMT XR-50 CCD was used to digitize images.

**Contraction analysis**. CM contraction kinetics were assessed using that disturbance in calcium oscillation patterns, observed using Ca2+-sensitive fluorescent dyes which can be utilized to measure effects on CM beat rate and pattern[59,60]. Calcium oscillation patterns were used to measure effects on CMs beat rate and contraction using custom module in MetaXpress. Each trace represents the contraction pattern from one representative cell. Contraction intensities from each siRNA-treated cell were normalized to control. CM contraction was assessed in two independent experiments.

**Treatment with PPARα agonist and PGC1/PPARα targets KD**. Day 20 hiPSC-CMs were dissociated with TrypLE Select 10X for up to 10 min. TrypLE was neutralized with RPMI supplemented with 10% FBS. Cells were resuspended in RPMI with 2% KOSR (Gibco) and 2% B27 50X (Life Technologies) supplemented with 2 μM Thiazovivin and 10 μM PPARα agonist. Next, each well of a Matrigel-coated 384-well plate was seeded with 6000 day 20 hiPSC-CMs. Ten micromolar PPARα agonist was refreshed every other day for 11 days.

To assess the role of PGC1/PPARα targets on calcium handling, day 20 hiPSC-CMs were transfected with siRNAs directed against 148 genes PGC1/PPARα targets using lipofectamine RNAi Max (ThermoFisher). PPARα agonist concentration (10 μM) was maintained constant for 11 days. To insure KD efficiency, hiPSC-CMs were transfected with siRNAs two additional times at day 24 and 28 of differentiation. Each siRNA was transfected in one-plicate for the primary screen (148 siRNAs) and in quadruplicate for the validation screen (50 siRNAs).

**Calcium assay in hiPSC-CMs**. Calcium assay is performed using labeling protocol as described[61]. After 11-day treatment with PPARα agonist and/or PGC1/PPARα target-directed siRNAs, 50 μL of media was removed and replaced in each well by a 2× calcium dye solution consisting in Fluo-4 NW dye (Invitrogen), 1.25 mM Probenecid F-127 (Invitrogen) and 100 μg/mL Hoescht 33258 (diluted in water, ThermoFisher) diluted in warm Tyrode's solution (Sigma). The plate was placed back in the 37 °C 5% CO2 incubator for 45 min. After incubation time, cells were washed four times with fresh pre-warmed Tyrode's solution by removing 50 μL of media and adding 50 μL of Tyrode's solution using a 16 channel pipette. hiPSC-CMs were then automatically imaged with ImageXpress Micro XLS microscope (Molecular Devices) at an acquisition frequency of 100 Hz for a duration of 5 s with an excitation wavelength of 485/20 nm and emission filter 525/30 nm. A single image of Hoescht was acquired before the time series. Fluorescence over time quantification and trace analysis were automatically quantified using custom software packages developed by Molecular Devices and Colas lab.

**Statistical analysis**. Population distribution of control and PPARα-treated hiPSC-CMs was generated with GraphPad Prism software (2019) using nonlinear regression. Unpaired nonparametric Kolmogorov-Smirnov test was used to compare each treated condition to control using CTD75 of every measured cell. To determine any statistical significance between experimental and control groups, we calculated two-sided p-values with Student's t-test using GraphPad Prism software. The images shown were representative and reproducible. scRNA-seq used multiple mice. The Mann-Whitney-Wilcoxon nonparametric test was used when distributions were not normal. The box-and-whiskers plot represents the maxima, 75th percentile, median, 25th percentile, and minima.

**Reporting summary**. Further information on research design is available in the Nature Research Reporting Summary linked to this article.

## Data availability

Sequencing data is accessible through GEO Accession GSE165917. Bulk RNA-seq data can be found at these GEO Accession numbers: GSE64403, GSE47948, GSE95762, GSE79883. Supplementary data can be found in the supplementary data tables. Source data are provided with this paper.

## Code availability

Custom scripts used to analyze data and generate figures can be found at https://github.com/smurph50/Cardiomyocyte-maturation-scRNAseq.

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

## Acknowledgements

We thank Deepthi Ashok and Dr. Brian O'Rourke for helping mitochondria assays, Danielle Rigau for animal work assistance, George McNamara for confocal imaging assistance, Michael Delannoy for electron microscopy assistance, and Dr. Daniel Kelly for providing technical assistance for ChIP. We thank Dr. Leslie Tung for helpful discussions and critical feedback. This work was supported by grants from NIH, MSCRF, AHA, and JHU TMTM.

## Author contributions

S.M. and M.M. designed and carried out this work. Suraj K. helped with LP-FACS and scRNA-seq analysis. A.K. and A.C. designed and performed high-throughput PSC-CM assays. B.L., D.K., and D.A.K. provided expertise in contractility assays. E.T. designed and performed hPSC-CM analysis. S.P. and S.A. performed traction microscopy. Sandeep K. performed animal work and image analysis. P.A. helped with designing in vitro screen. R.Z. helped with scRNA-seq data analysis. H.U. provided conceptual input. C.K. designed and supervised this work and wrote the manuscript with S.M.

## Competing interests

The authors declare no competing interests.
