## [Peer Review File · Nature Communications]

Reviewers' comments:

Reviewer #1 (Remarks to the Author):

Title: PGC1/PPAR Drive Cardiomyocyte Maturation through Regulation of Yap1 and SF3B2

In this manuscript, the author identified the factors and the candidate signaling pathway for the maturation of human pluripotent stem cell-derived cardiomyocytes through updated cardiomyocytes single cell RNA sequencing method with mouse cardiomyocytes in a timely manner. The author figured out that PGC1/PPAR signaling is mainly involved in the cardiomyocytes maturation process through the scRNA-seq analysis of the primary mouse cardiomyocytes and they confirmed that activation of the signaling pathway leads to hPSC-CM maturation. The manuscript is well organized and the strategy to prove the concept is straightforward. Some changes will improve the manuscript

Major Comments:

1. Regarding figure 2C, the author did not mention what the first ranked gene was (left top dot) and why the first ranked factor was neglected for the further analysis.
2. Regarding line 148 and figure 2B, it appears that the selection of PPARG makes more sense than PPARG based on the heat map data. The rationale to choose the PGC1A and PPARG versus PPARG is unclear.
3. In figure S2A, B and C, a scheme for PGC1 beta deletion and confirmation data are missing.
4. In figure S2E-L, statistical analysis are missing.
5. Regarding figure 4G, I am wondering whether comparing the differentially expressed gene with ChIPseq target genes is appropriate to find out the downstream targets of the signaling because PGC1-regulated genes do not represent only maturation. In addition, ChIP-seq was performed with whole heart tissues, not cardiomyocytes only.
6. In figure 5A, low magnified images are required to confirm homogeneous maturation.
7. For the maturation analysis, more analysis would help such as the electrophysiological studies and sarcomere length measurement.

Minor Comments

1. Over the manuscript, some minor changes are required such as the order of the figure number (figure 2C comes first in the manuscripts than 2B) or capitalized "D" in line 147.
2. In figure 1B, C, F and 4A, color changes of each dot is recommended to present a segregation of the cell population.
3. Regarding single cell RNA sequencing, the number of cells and reads were not described.
4. Regarding figure 5, a timetable which include the differentiation date and the period of chemical treatment is required for better understanding of the manuscript.

Reviewer #2 (Remarks to the Author):

This manuscript by Kwon and colleagues describes the role of PGC1/PPAR pathway to drive maturation of cardiomyocyte by regulating Yap1 and SF3B2. The authors employed a novel large-particle sorting approach to analyze single cardiomyocytes (CMs) from neonatal and adult mouse hearts. Neonatal CMs exhibit heterogeneity of size and transcriptome and continuously changed until adulthood. Analysis of gene regulatory network followed by gene loss-of-function studies show that PGC1 signaling mediated by Yap1 and SF3B2 is required for transcriptional maturation of CMs in vivo. This study highlights a previously unrecognized role of Yap1 and SF3B2 in CM maturation and demonstrated the power and utility of single cell transcriptional profiling of adult CMs. Overall, the findings are quite exciting and timely given the intense interest in CM maturation and single cell transcriptome analysis currently. Some minor points are listed below to improve the data presentation and robustness of the conclusions reached.

1. There is general agreement that UMAP represents a more accurate representation of the transcriptome relationship between two cells in 2D than tSNE. The authors are encouraged to convert Figure 1B from tSNE plot to UMAP plot.
2. Figure 1D - it would be helpful to give an explanation why there is more heterogeneity in maturation score for CMs at day P28 than CMs at day P14. In particular, why are there more cells that exhibit lower mature score at P28 than at P14.
3. Figure 2B (line 47) and 2C (line 44) are described in reverse sequence
4. Beyond changes in transcriptome, cell size, sarcomere distance, CM maturation is also characterized by the presence of t-tubule and conversion of energy substrate utilization from glucose to fatty acid. It would be good to show the presence of t-tubule and conversion to fatty acid consumption in study of human iPSC-CM treated with PGC1a or PPARa agonist in Figure 5A.
5. Along with point #4, it would be helpful if the authors can add a few sentences in the Discussion section to describe mechanistically how transcriptional changes mediated by PGC1a/PPARa can lead, eventually, to improvement in CM maturation phenotype. Highlighting some of the down-stream targets of PGC1a/PPARa (e.g. mitochondrial genes involved in ATP production) that have been described to regulate fatty acid oxidation and, indirectly, CM maturation would be helpful.
6. The font sizes for most of the letters within each Figure (e.g. Fig 1B,C,D,E,F; Fig 2A,B; Fig 4A,B,E, etc) and axis labels are quite small. It would help to increase the sizes of these letters for general audience, especially if these figures are reduced in size in print form.

Reviewer #3 (Remarks to the Author):

Review of NCOMMS-20-02357-T

PGC1/PPAR Drive Cardiomyocyte Maturation through Regulation of Yap1 and SF3B2

Murphy et al. describe the role of PGC1/PPAR factors in the regulation of cardiomyocyte maturation. The authors used large particle sorting of mouse cardiomyocytes isolated at postnatal days 0, 7, 14, 21, and 28 to perform single cell RNA-seq. This revealed a unidirectional developmental trajectory, with substantial heterogeneity at individual timepoints. Upstream regulatory analysis predicts that the PPAR cofactors PGC-1 α/β are important transcriptional regulators of the maturation process, and so the authors examine how maturation is affected by loss and gain-of-function approaches in vivo and in stem-cell derived cardiomyocytes. The authors show PGC-1 α/β perturbation does indeed affect cardiomyocyte maturation state, and identify potential directly regulated downstream effectors including Yap1, which is necessary for the PGC-1 α/β maturational effect.

The manuscript includes two important technical innovations. First, the large particle sorting of mouse cardiomyocytes will be useful for isolation of adult cardiomyocytes, which are problematic for other types of cell sorters. The large particle sorted cardiomyocytes appear to be suitable for physiology experiments, in addition to scRNA-seq. In contrast, conventional FACS cells are extensively damaged and too large for 10X Chromium scRNA-seq (although their nuclei can be used for snRNA-seq). On the other hand, this method of scRNA-seq is far lower throughput. Second, the manuscript uses relatively high throughput physiology measurements to screen a set of siRNAs for those that alter the calcium transient potential.

The experiments are well conducted and extensive. The authors identified PGC1/PPAR signaling as important for cardiomyocyte maturation. Prior work, including a previous study by this group (Cell

Reports, 2015), pointed to PGC1/PPAR in cardiomyocyte maturation, and these genes have been extensively studied in the induction of fatty acid metabolism and mitochondrial biogenesis, key elements of cardiomyocyte maturation. However, a relatively new element from this study is the finding that PGC1/PPAR have a broader role in regulating non-metabolic aspects of cardiomyocyte maturation. This was nicely supported by mosaic inactivation of PGC1a/b and demonstration of altered cell size, contraction, and calcium transients.

Major Points

1. scRNA-seq: it is not clear how many cells from each stage were profiled, and at what depth. Were enough cells analyzed to yield informative clustering in the t-SNE plots or might those be misleading due to insufficient cell density? Although the authors propose that the scRNA-seq data is a valuable resource, its value is limited by the low cell number. On the other hand, perhaps these libraries are more suitable to deep sequencing than 10X data and therefore could provide unique information by permitting deeper transcriptome interrogation? This is hard to evaluate based on the current data presentation.
2. Cardiomyocytes are known to be heterogeneous. For example, heterogeneity between compact and trabecular CMs, or between CMs near the endocardial vs epicardial surface. The authors seem to equate transcriptome heterogeneity to heterogeneity of maturation without accounting for how these other types of heterogeneity may impact transcriptomic assessment of maturation.
3. The cell size measurements have substantial disagreement across figures. For example, in 3C the median P28 cell is 1280 μm^2 , but in Fig. 5E, it is 2120 μm^2 .
4. The differences in maturation scores for Figure 4B are modest, and there are no statistics on the plot. Are any of the differences significant?
5. There is very little information for the ChIP-seq experiment. How many replicates for each factor? How many peaks were called for each factor? How well did replicates correlate? How much overlap was observed between PGC1a and PPARa results?
6. The authors use two criteria to identify genes "directly" regulated by PGC1a – differentially expressed in mosaic KO, and occupied by PGC1a (what was the association rule to link ChIP peaks to genes?). This is supportive of direct regulation, but not all binding leads to functional gene regulation. Use of reporter assays of important putative regulatory elements (e.g. near YAP or SF3B2) with or without target motif mutation would provide additional supportive data.
7. YAP1 knockout had a relatively weak effect of cardiomyocyte growth – cardiomyocyte size was 20% less in YAP1 knockout myocytes compared to controls, whereas cardiomyocytes increase in size by 220% during maturation, which was reduced to 128% by PGC1 knockout. This suggests that YAP is a minor contributor to the effect of PGC1. The authors cite another study (von Gise, PNAS, 2012) which concluded that YAP did not significantly alter maturational growth of cardiomyocytes. Sadoshima's group previously proposed that YAP was required for cardiomyocyte hypertrophy (Del Re, JBC, 2013). This study was not cited.
The data showing Yap1 to be a pro-maturation factor is interesting but contrary to expectations as Yap1 is a strong cell-cycle activator while maturing CMs exit the cell-cycle. Some discussion of this puzzle, and perhaps of the maturational expression profile of Yap1 would be beneficial.
8. siRNAs were used to screen genes downstream of PGC1a to find those that mediate shortened Ca transient duration in the presence of PGC1a agonist. Was CaTD or action potential duration increased in murine PGC1a/b mutant cells?
9. The screen identified three that prolong calcium transient duration. RNA splicing factor SF3B2 is particularly interesting and one of the more novel findings from this study. However, this portion of the study is largely relegated to supplemental data and is underdeveloped. What other maturational parameters do these factors regulate? Are these factors required for maturation of murine cardiomyocytes? Is splicing altered in murine PGC-1a/ β KO cardiomyocytes?

Minor Points

1. In Figure 1E, cell volumes appear to be extrapolated from some a parameter on the sorting instrument. What is ToF? How does this assessment of cell size compare to manual size measurement? The trajectory of cell size does not appear similar to Fig. 3C.

2. In Fig. 2A, C, to what extent is the high ranking of PGC1a and PPAR driven by the changes in metabolic/mitochondrial genes? In later figures the authors argue for wider roles of these factors in CM maturation, and it would be interesting to know if non-metabolic/mitochondrial signatures of PGC1a or PPAR contribute to their high ranking for maturation regulators.
3. In Figure 2D, are these correlations significant? What is the gray region around the smooth line of fit?
4. Figure 3D the PGC1 mutant CMs have a longer sarcomere length, which is the opposite of the expected result if the cells are immature.
5. In Figure 3G, what are the individual data points? The legend says n=9-12, but these individual data points do not seem to represent 9-12 different recordings.
6. In Figure 4A the colors used for mutant and control are too similar. Can mutants use open symbols or symbols with a different shape.
7. Figure 4F, the motif logos are displayed such that the letter height does not correspond to information content. This makes the logos hard to interpret. In Homer, use option -bits. Many nuclear receptors share motifs and it can be difficult to assign specific NRs to enriched motifs.
8. In Figure 5, PGC1a agonists were used, but the authors show that PGC1 and PPAR levels are low in iPSC-CMs. Do these agonists upregulate PGC1 and PPAR? Otherwise it is hard to understand how activation of such lowly expressed genes induce the phenotypic changes. Are the changes abolished by PGC1 knockout or might these agonists have off-target activity?
9. Line 244 references Figure 2E, which is absent.

Reviewer #4 (Remarks to the Author):

This manuscript reports additional information related to the role of the transcriptional coactivator PGC1a/b in cardiomyocyte maturation with implications in heart function in healthy and disease states. The authors developed and applied an experimental approach to perform large-particle sorting and analyzed RNA profiles in single myocytes from neonatal to adult stages. As expected, based on previous studies, they found that one of the major pathways associated with cardiomyocyte maturation is the transcriptional coactivator PGC1a/b and the downstream metabolic/mitochondrial network. The authors generated mosaic heart studies in vivo to delete PGC1a/b in a percentage of cardiomyocytes that were analyzed for gene expression analysis. Again, as predicted a series of genes linked to metabolism, calcium handling and size regulators were identified. A specific emphasis and analysis were pursued to YAP and SF3B2 splicing factor (it is known that PGC1a associates with splicing proteins) to define potential requirements for cell size. Overall, the manuscript is well executed and largely confirmed previous studies on PGC1a and heart biology and development. A major concern of this manuscript relates to 1) the novelty of the studies, and 2) incremental knowledge on PGC1a function in heart with the inclusion of YAP and SF32B, and 3) problematic use of pharmacological activators of PGC1a and PPARa.

- 1- Although the authors use an elegant series of experiments using scRNA seq in different cardiomyocytes population, the outcomes of these experiments are largely confirmative of previous studies.
- 2- A major point the authors highlight is the cell size- mainly related to YAP- There are important caveats on these studies. First, manipulations of YAP in these studies by LOF and GOF are not sufficiently developed; but more importantly the connection between PGC1 and YAP is problematic- It is very likely that these effects are indirect through cell cell interactions created with the mosaic model. In addition, mTOR have been linked to PGC1 and also, differentiation or maturation in this case through PGC1a/b includes mitochondrial mass that might account for the cell size; or even the calcium handling pathways are associated with cell size.
- 3- Related to the previous point, the authors also include the SF32B as a novel mediator of PGC1a on cardiomyocyte maturation, through potential splicing. This could be a novel addition, but it will require large amount of experiments to characterize this factor.

- 4- The use of PGC1a activator, is very, very problematic there is no experimental evidence that this is an specific activator of this protein and experiments with PGC1a/b mutants should define the specificity. Similar critiques apply to the PPARa-
- 5- The PPARa is also problematic because it only defines a fraction of PGC1a/b metabolic control linked to fatty acid oxidation, which is largely unexplored in this manuscript.

Response to critiques

We greatly appreciate all the reviewers' critical and thoughtful comments and the editor for inviting a revised manuscript. Unfortunately, the COVID-19 pandemic started when we received the comments, and we have faced significant challenges in performing wet lab experiments. Nevertheless, we have tried our best to address all questions raised by the reviewers within our capability. Our point-by-point responses to the reviewers' comments are as follows:

Reviewer #1 (Remarks to the Author):

In this manuscript, the author identified the factors and the candidate signaling pathway for the maturation of human pluripotent stem cell-derived cardiomyocytes through updated cardiomyocytes single cell RNA sequencing method with mouse cardiomyocytes in a timely manner. The author figured out that PGC1/PPAR signaling is mainly involved in the cardiomyocytes maturation process through the scRNA-seq analysis of the primary mouse cardiomyocytes and they confirmed that activation of the signaling pathway leads to hPSC-CM maturation. The manuscript is well organized and the strategy to prove the concept is straightforward. Some changes will improve the manuscript

Major Comments:

1. Regarding figure 2C, the author did not mention what the first ranked gene was (left top dot) and why the first ranked factor was neglected for the further analysis.

: The first gene is *NFKBIA*. While *NFKBIA* may play an important role in CM development, we found that its expression levels decrease over time (see figure below). Thus, we chose to focus on the second ranked gene PGC1, whose levels continue to increase during postnatal CM maturation (Fig. 2d).

2. Regarding line 148 and figure 2B, it appears that the selection of PPARG makes more sense than PPARA based on the heat map data. The rationale to choose the PGC1A and PPARA versus PPARG is unclear.

: We agree with the reviewer that *PPARg* is expressed slightly lower than *PPARa* in PSC-CMs (Fig. 2c), and in fact have conducted experiments with *PPARg* in mESC-CMs (not shown in the manuscript). In brief, we generated a Myom2-RFP ESC line, where RFP levels are increased during CM maturation (Chanthra et al., 2020; Stainer et al., 1998). When tested with *PPARa/g/d*

ligands, we observe a notable increase in RFP levels with the PPAR α ligand-treated PSC-CMs, but not in PPAR γ ligand-treated PSC-CMs (see below). Further, while both PPAR α and PPAR γ isoforms share structural and functional features, PPAR α is known to be important for myofibril structure and contractility in CMs (Watanabe et al., 2000; Bugge et al., 2010), in line with the postnatal role of PGC1 that we have found in the current study (Fig. 4e). Therefore, we chose PPAR α over PPAR γ in our study.

- Watanabe, K. *et al.* Constitutive regulation of cardiac fatty acid metabolism through peroxisome proliferator-activated receptor alpha associated with age-dependent cardiac toxicity. *J Biol Chem* **275**, 22293-22299, doi:10.1074/jbc.M000248200 (2000).
- Bugge, A. & Mandrup, S. Molecular Mechanisms and Genome-Wide Aspects of PPAR Subtype Specific Transactivation. *PPAR Res* **2010**, doi:10.1155/2010/169506 (2010).
- Chanthra N, Abe T, Miyamoto M, et al. A Novel Fluorescent Reporter System Identifies Laminin-511/521 as Potent Regulators of Cardiomyocyte Maturation. *Sci Rep.* 2020;10(1):4249. Published 2020 Mar 6. doi:10.1038/s41598-020-61163-3
- Steiner, F., Weber, K. & Furst, D. O. Structure and expression of the gene encoding murine M-protein, a sarcomere-specific member of the immunoglobulin superfamily. *Genomics* **49**, 83-95, doi:10.1006/geno.1998.5220 (1998).

3. In figure S2A, B and C, a scheme for PGC1 beta deletion and confirmation data are missing.

: The schematic and data for PGC1b deletion are now included in Figure S2d, e. We also increased the number of replicates for PGC1a deletion (200 CMs from 5 mice).

4. In figure S2E-L, statistical analysis are missing.

: We have now added statistical analysis.

5. Regarding figure 4G, I am wondering whether comparing the differentially expressed gene with ChIPseq target genes is appropriate to find out the downstream targets of the signaling because PGC1-regulated genes do not represent only maturation. In addition, ChIP-seq was performed with whole heart tissues, not cardiomyocytes only.

: We agree with this reviewer that we used entire hearts for the analysis including non-CMs and thus, these results are not sufficient to conclude that the identified genes are directly regulated by PGC1/PPARa signaling for CM maturation. We acknowledge that this is a valid point. Since our manuscript is focused on single cell analysis, we determined that the bulk ChIP-seq analysis would not be necessary and rather diffuses the focus. Thus, we have excluded the data in the revised manuscript. Instead, we have performed more specific ChIP-qPCR and showed that PGC1/PPARa are associated with the promoters of Yap1 and SF3B2—factors that mediate PGC1/PPARa signaling for hypertrophy and contractility development. We have included the new data (Fig. S4 c, d) and changed the text accordingly.

6. In figure 5A, low magnified images are required to confirm homogeneous maturation. For the maturation analysis, more analysis would help such as the electrophysiological studies and sarcomere length measurement.

: Originally, we replated PSC-CMs at low density to obtain single CM images for quantification. In order to obtain low magnified images, we planned to replate PSC-CMs at a higher density, but unfortunately, our imaging core with the confocal microscope was closed due to the COVID-19 pandemic. Similarly, we planned to perform the electrophysiological studies through a collaboration with Dr. Leslie Tung's lab at Hopkins, but have been forced to delay this plan for the same reason. We hope that reviewers understand the limitation. We have analyzed key features of CM maturation, including cellular hypertrophy, contractility, and calcium handling development (Figs 5–7), which we believe is sufficient to conclude that PGC1 signaling promotes PSC-CM maturation.

Minor Comments

1. Over the manuscript, some minor changes are required such as the order of the figure number (figure 2C comes first in the manuscripts than 2B) or capitalized “D” in line 147.

: Corrected accordingly.

2. In figure 1B, C, F and 4A, color changes of each dot is recommended to present a segregation of the cell population. - ????

: These colors are associated with each time-point of isolation and used by numerous scRNA-seq papers. We hope this is acceptable.

3. Regarding single cell RNA sequencing, the number of cells and reads were not described.

: We sequenced 300 control and 328 PGC1 cmKO CMs that passed our quality cutoff. Here is a histogram of UMIs per cell. In our experimental design, we allotted 1 million reads per cell since the Illumina NextSeq has 400 million reads per lane and we ran two lanes.

4. Regarding figure 5, a timetable which include the differentiation date and the period of chemical treatment is required for better understanding of the manuscript.

: We have added a timeline showing the differentiation and treatment in Fig. 5a.

Reviewer #2 (Remarks to the Author):

This manuscript by Kwon and colleagues describes the role of PGC1/PPAR pathway to drive maturation of cardiomyocyte by regulating Yap1 and SF3B2. The authors employed a novel large-particle sorting approach to analyze single cardiomyocytes (CMs) from neonatal and adult mouse hearts. Neonatal CMs exhibit heterogeneity of size and transcriptome and continuously changed until adulthood. Analysis of gene regulatory network followed by gene loss-of-function studies show that PGC1 signaling mediated by Yap1 and SF3B2 is required for transcriptional maturation of CMs in vivo. This study highlights a previously unrecognized role of Yap1 and SF3B2 in CM maturation and demonstrated the power and utility of single cell transcriptional profiling of adult CMs. Overall, the findings are quite exciting and timely given the intense interest in CM maturation and single cell transcriptome analysis currently. Some minor points are listed below to improve the data presentation and robustness of the conclusions reached.

1. There is general agreement that UMAP represents a more accurate representation of the transcriptome relationship between two cells in 2D than tSNE. The authors are encouraged to convert Figure 1B from tSNE plot to UMAP plot.

: Thank you for the comment. We remade the plot with UMAP using the Seurat R package and added it to Fig. S1a. We initially chose tSNE because it has been shown that tSNE and UMAP have similar preservation of global structure when properly optimized (Kobak and Linderman, 2019), with similar distortion (Cooley et al., 2019). In our case, we notice a strong concordance between tSNE and DDRTree (Fig. 1c), and therefore favored tSNE over UMAP in our study.

- Kobak D, Linderman GC. UMAP does not preserve global structure any better than t-SNE when using the same initialization. *bioRxiv*, doi:10.1101/2019.12.19.877522 (2019).
- Cooley SM, Hamilton T, Deeds EJ, Ray CJ. A novel metric reveals previously unrecognized distortion in dimensionality reduction of scRNA-seq data. *bioRxiv*, doi:10.1101/689851 (2019).

2. Figure 1D - it would be helpful to give an explanation why there is more heterogeneity in maturation score for CMs at day P28 than CMs at day P14. In particular, why are there more cells that exhibit lower mature score at P28 than at P14.

: We expect some cells to have lower maturation score at any given time point. The maturation score plateaus between p14 and p28. Since we have more cells analyzed at p28, we are likely to see more outliers (very immature cells). This is supported by another study, where we used a larger number of CMs (Kannan et al., 2020).

- Kannan S, Farid M, Lin BI, Miyamoto M, Kwon C. (2020) Transcriptomic Entropy Quantifies Cardiomyocyte Maturation at Single Cell Level. *bioRxiv* 22632

3. Figure 2B (line 47) and 2C (line 44) are described in reverse sequence

: We have corrected this by switching the labels.

4. Beyond changes in transcriptome, cell size, sarcomere distance, CM maturation is also characterized by the presence of t-tubule and conversion of energy substrate utilization from glucose to fatty acid. It would be good to show the presence of t-tubule and conversion to fatty acid consumption in study of human iPSC-CM treated with PGC1a or PPARa agonist in Figure 5A.

: We agree that these would strengthen the manuscript. We have shown that contractile force, calcium handling, and metabolic capacity—key aspects of CM maturation—are significantly improved with PGC1/PPAR activation (Figs 5d, 6b, S3g). But unfortunately, we were unable to perform additional experiments due to the institutional regulation on wet bench work and core facilities. Nevertheless, we believe that the data we have makes a convincing case that PGC1 signaling promotes PSC-CM maturation.

5. Along with point #4, it would be helpful if the authors can add a few sentences in the Discussion section to describe mechanistically how transcriptional changes mediated by PGC1a/PPARa can lead, eventually, to improvement in CM maturation phenotype. Highlighting some of the down-stream targets of PGC1a/PPARa (e.g. mitochondrial genes involved in ATP production) that have been described to regulate fatty acid oxidation and, indirectly, CM maturation would be helpful.

: Thank you for this comment. As you know, FAO and OXPHOS are two key pathways used for energy metabolism, and our data show that PGC1 regulates OXPHOS, not FAO. We have added the following sentences in Discussion: “PGC1/PPAR signaling regulates numerous genes involved in mitochondria biogenesis and cellular metabolism. Curiously, fatty acid treatment was shown to enhance structural and functional maturation of PSC-CMs (Yang et al., 2019), similar to PGC1/PPAR activation. However, gene ontology analysis suggests that the underlying mechanisms may be different: while fatty acids regulate genes and pathways involved in fatty acid β -oxidation and lipid synthesis, PGC1 affects cardiac muscle development and oxidative phosphorylation. Nevertheless, these suggest the importance of cellular metabolism/energy production in PSC-CM maturation. It would be interesting to study if YAP1/SF3B2 influence genes involved in energy metabolism”.

- Yang, X. et al. Fatty acids enhance the maturation of cardiomyocytes derived from human pluripotent stem cells. *Stem Cell Rep.* 2019;13(4):657-668.

6. The font sizes for most of the letters within each Figure (e.g. Fig 1B,C,D,E,F; Fig 2A,B; Fig 4A,B,E, etc) and axis labels are quite small. It would help to increase the sizes of these letters for general audience, especially if these figures are reduced in size in print form.

: We have now increased the font sizes.

Reviewer #3 (Remarks to the Author):

Murphy et al. describe the role of PGC1/PPAR factors in the regulation of cardiomyocyte maturation. The authors used large particle sorting of mouse cardiomyocytes isolated at postnatal days 0, 7, 14, 21, and 28 to perform single cell RNA-seq. This revealed a unidirectional developmental trajectory, with substantial heterogeneity at individual timepoints. Upstream regulatory analysis predicts that the PPAR cofactors PGC-1 α/β are important transcriptional regulators of the maturation process, and so the authors examine how maturation is affected by loss and gain-of-function approaches in vivo and in stem-cell derived cardiomyocytes. The authors show PGC-1 α/β perturbation does indeed affect cardiomyocyte maturation state, and identify potential directly regulated downstream effectors including Yap1, which is necessary for the PGC-1 α/β maturational effect.

The manuscript includes two important technical innovations. First, the large particle sorting of mouse cardiomyocytes will be useful for isolation of adult cardiomyocytes, which are problematic for other types of cell sorters. The large particle sorted cardiomyocytes appear to be suitable for physiology experiments, in addition to scRNA-seq. In contrast, conventional FACS cells are extensively damaged and too large for 10X Chromium scRNA-seq (although their nuclei can be used for snRNA-seq). On the other hand, this method of scRNA-seq is far lower throughput. Second, the manuscript uses relatively high throughput physiology measurements to screen a set of siRNAs for those that alter the calcium transient potential.

The experiments are well conducted and extensive. The authors identified PGC1/PPAR signaling as important for cardiomyocyte maturation. Prior work, including a previous study by this group (Cell Reports, 2015), pointed to PGC1/PPAR in cardiomyocyte maturation, and these genes have been extensively studied in the induction of fatty acid metabolism and mitochondrial biogenesis, key elements of cardiomyocyte maturation. However, a relatively new element from this study is the finding that PGC1/PPAR have a broader role in regulating non-metabolic aspects of cardiomyocyte maturation. This was nicely supported by mosaic inactivation of PGC1a/b and demonstration of altered cell size, contraction, and calcium transients.

Major Points

1. scRNA-seq: it is not clear how many cells from each stage were profiled, and at what depth. Were enough cells analyzed to yield informative clustering in the t-SNE plots or might those be misleading due to insufficient cell density? Although the authors propose that the scRNA-seq data is a valuable resource, its value is limited by the low cell number. On the other hand, perhaps these libraries are more suitable to deep sequencing than 10X data and therefore

could provide unique information by permitting deeper transcriptome interrogation? This is hard to evaluate based on the current data presentation.

: We appreciate this comment. We used 300 cells for the control CM trajectory and 328 for the mutant CM trajectory, and are already sequencing them at *one order of magnitude higher depth* than typical 10X libraries. This information and average depth have been added to the Fig. 1 legend and text. This higher depth is particularly important in our study because analyzing CM maturation requires high-depth sequencing due to their gradual transcriptomic changes. Bagnoli et al. have also shown that mcSCR-seq, which we used to prepare these libraries, is one of the most sensitive UMI-based techniques available, and thus we get much higher capture of lowly expressed genes including transcriptional regulators. This task is difficult to achieve with droplet methods.

The trade-off of our method is limited numbers of cells, as pointed out by the reviewer. Our newly developed LP-FACS method is 96 well plate-based and thus is not as high throughput as 10X, but we analyzed at least 50 CMs per time points to capture major trends over pseudotime. We have recapitulated these results with a lot more cells and time points in another study (Kannan et al., 2020), demonstrating reproducibility. Further, we want to emphasize that mature CMs are too large for droplet-based method (such as 10X) to process, and nuclear RNA seq does not provide the necessary depth.

- Bagnoli et al. Sensitive and powerful single-cell RNA sequencing using mcSCR-seq. *Nat Comms* **9**, 2937 (2018).
- Kannan S, Farid M, Lin BI, Miyamoto M, Kwon C. Transcriptomic Entropy Quantifies Cardiomyocyte Maturation at Single Cell Level. *bioRxiv* 22632 (2020).

2. Cardiomyocytes are known to be heterogeneous. For example, heterogeneity between compact and trabecular CMs, or between CMs near the endocardial vs epicardial surface. The authors seem to equate transcriptome heterogeneity to heterogeneity of maturation without accounting for how these other types of heterogeneity may impact transcriptomic assessment of maturation.

: We agree with the reviewer that there are other types of heterogeneity within CMs, and even though we focused our isolation on free wall ventricular CMs, we will inevitably have these other types of heterogeneity. However, in constructing the trajectories in Fig 1C, we used genes that were differentially expressed between P0 and P28 as the ordering list, allowing us to prioritize differences that happen over time (e.g. maturation). Moreover, the general concordance of maturation score with timepoint suggests that we successfully capture heterogeneity of the maturation process. It is very possible that heterogeneity of the maturation process *is* spatially mediated - but that is beyond the scope of this study and would require more defined spatial analysis.

3. The cell size measurements have substantial disagreement across figures. For example, in 3C the median P28 cell is 1280 μm^2 , but in Fig. 5E, it is 2120 μm^2 .

: We believe that the disagreement can be due to differences in mouse background/strain and age. PGC1a/b mice were generated with 129/SvJ strain while Yap1 mice were generated with 129P2/OlaHsd (Lin et al., 2004; Lai et al., 2008; Zhang et al., 2010). The Yap1 mice were P33, while the PGC1 mice were P28. We have re-quantified the images using imageJ to check for errors, but we concluded that this is likely a biological difference, but does not affect the interpretation of our data.

- Lin J; Wu PH; Tarr PT; Lindenberg KS; St-Pierre J; Zhang CY; Mootha VK; Jager S; Vianna CR; Reznick RM; Cui L; Manieri M; Donovan MX; Wu Z; Cooper MP; Fan MC; Rohas LM; Zavacki AM; Cinti S; Shulman GI; Lowell BB; Krainc D; Spiegelman BM. 2004. Defects in adaptive energy metabolism with CNS-linked hyperactivity in PGC-1 α null mice. *Cell* 119(1):121-35
- Lai L; Leone TC; Zechner C; Schaeffer PJ; Kelly SM; Flanagan DP; Medeiros DM; Kovacs A; Kelly DP. 2008. Transcriptional coactivators PGC-1 α and PGC-1 β control overlapping programs required for perinatal maturation of the heart. *Genes Dev* 22(14):1948-61
- Zhang N; Bai H; David KK; Dong J; Zheng Y; Cai J; Giovannini M; Liu P; Anders RA; Pan D. 2010. The Merlin/NF2 tumor suppressor functions through the YAP oncoprotein to regulate tissue homeostasis in mammals. *Dev Cell* 19(1):27-38

4. The differences in maturation scores for Figure 4B are modest, and there are no statistics on the plot. Are any of the differences significant?

: We performed statistical analysis and found that all of the differences are significant.

5. There is very little information for the ChIP-seq experiment. How many replicates for each factor? How many peaks were called for each factor? How well did replicates correlate? How much overlap was observed between PGC1 α and PPAR α results?

: After carefully revisiting and discussing the experiment, we realized that the bulk ChIP-seq data, obtained from entire hearts containing various non-CMs, would not be sufficient to conclude that the identified genes are directly regulated by PGC1/PPAR α signaling for CM maturation. We also think that the bulk ChIP-seq analysis may diffuse the focus of this manuscript—single cell analysis of postnatal CMs during maturation. For these reasons (and with the recommendations of other reviewers), we have removed the data in the revised manuscript. Instead, we have performed more specific ChIP-qPCR to show that PGC1/PPAR α are associated with the promoters of *Yap1* and *SF3B2*—factors that mediate PGC1/PPAR α signaling for hypertrophy and contractility development. We have included the new data in Fig. S4 and changed the text accordingly.

6. The authors use two criteria to identify genes “directly” regulated by PGC1a – differentially expressed in mosaic KO, and occupied by PGC1a (what was the association rule to link ChIP peaks to genes?). This is supportive of direct regulation, but not all binding leads to functional gene regulation. Use of reporter assays of important putative regulatory elements (e.g. near YAP or SF3B2) with or without target motif mutation would provide additional supportive data.

: We appreciate this point. As mentioned above, we performed more specific ChIP-qPCR and toned down the interaction to “physical association” in their promoter regions.

7. YAP1 knockout had a relatively weak effect of cardiomyocyte growth – cardiomyocyte size was 20% less in YAP1 knockout myocytes compared to controls, whereas cardiomyocytes increase in size by 220% during maturation, which was reduced to 128% by PGC1 knockout. This suggests that YAP is a minor contributor to the effect of PGC1. The authors cite another study (von Gise, PNAS, 2012) which concluded that YAP did not significantly alter maturational growth of cardiomyocytes. Sadoshima’s group previously proposed that YAP was required for cardiomyocyte hypertrophy (Del Re, JBC, 2013). This study was not cited. The data showing Yap1 to be a pro-maturation factor is interesting but contrary to expectations as Yap1 is a strong cell-cycle activator while maturing CMs exit the cell-cycle. Some discussion of this puzzle, and perhaps of the maturational expression profile of Yap1 would be beneficial.

: We agree with the reviewer that PGC1-deleted CMs show more pronounced effects on cell size than Yap1-deleted CMs. However, suppressing YAP1 activity completely abolished PGC1-mediated hypertrophy in PSC-CMs (Fig. 5g), suggesting its permissive role in this context.

We thank the reviewer for bringing up the work by Dr. Sadoshima’s group, where they showed an instructive role of Yap1 for neonatal CM growth. We have now cited the paper (Del Re et al., 2013).

We also acknowledge that Yap1 is a potent cell-cycle regulator, which appears to be in disagreement with its role in CM maturation. However, recent studies suggest that Yap1 directly regulates cell volume regardless of cell division or mTOR signaling via regulating cytoplasmic pressure (Perez-Gonzalez et al., 2019). Considering cellular hypertrophy is a key aspect of CM maturation, this suggests that Yap1 may promote maturation independently of proliferation. Supporting this idea, our single CM expression profile shows that *Yap1* levels is maintained during maturation (Fig. 1f).

- Perez-Gonzalez, N.A., et al. YAP and TAZ regulate cell volume. *J Cell Biol.* 218.10 (2019): 3472-3488.
- Del Re, D.P. et al. Yes-associated protein isoform 1 (Yap1) promotes cardiomyocyte survival and growth to protect against myocardial ischemic injury. *J Biol Chem.* 288(6): 3977-3988

8. siRNAs were used to screen genes downstream of PGC1a to find those that mediate shortened Ca transient duration in the presence of PGC1a agonist. Was CaTD or action potential duration increased in murine PGC1a/b mutant cells?

: That is correct. The calcium transient duration was significantly longer in PGC1-cmKO cells compared to control cells (Figure S2L). This is consistent with the impaired CaTD duration observed in PSC-CMs with siRNAs against *SF3B2/SAP18*, *TIMM50*, and *STRIP1* (Fig. 7).

9. The screen identified three that prolong calcium transient duration. RNA splicing factor SF3B2 is particularly interesting and one of the more novel findings from this study. However, this portion of the study is largely relegated to supplemental data and is underdeveloped. What other maturational parameters do these factors regulate? Are these factors required for maturation of murine cardiomyocytes? Is splicing altered in murine PGC-1a/β KO cardiomyocytes?

: We have now moved the key data to Fig. 7. In addition, we analyzed contractility of PSC-CMs deficient of the four identified factors and quantified their knockdown efficiencies (Figs. 7J and S5).

While we agree with the reviewer that *SF3B2* is indeed a novel gene whose function has not been explored, investigating its additional function and mechanism would be beyond the scope of this study.

Minor Points

1. In Figure 1E, cell volumes appear to be extrapolated from some a parameter on the sorting instrument. What is ToF? How does this assessment of cell size compare to manual size measurement? The trajectory of cell size does not appear similar to Fig. 3C.

: Time of flight (ToF) tracks the time it takes the cell to get from one measured point of the flow cell to another; larger cells will take longer and thus have longer time of flight. Thus, ToF is a good estimate of volume rather than the exact concordance. Fig. 3C is measuring the cell area, and this explains differences between the two. We have added the definition of ToF to the figure legend.

2. In Fig. 2A, C, to what extent is the high ranking of PGC1a and PPAR driven by the changes in metabolic/mitochondrial genes? In later figures the authors argue for wider roles of these factors in CM maturation, and it would be interesting to know if non-metabolic/mitochondrial signatures of PGC1a or PPAR contribute to their high ranking for maturation regulators.

: We used IPA, which is based on published literatures describing interactions. Thus, it is likely that the abundance of metabolic and mitochondrial genes is the primary reason for PGC1/PPARs being predicted as top regulators.

3. In Figure 2D, are these correlations significant? What is the gray region around the smooth line of fit?

: Both PPARa and PGC1a are significantly differentially expressed. The Monocle R package was used to generate a list of differentially expressed genes. The gray region indicates the 95% confidence interval.

4. Figure 3D the PGC1 mutant CMs have a longer sarcomere length, which is the opposite of the expected result if the cells are immature.

: The image shows p28 PGC1-deficient CMs. The presence of wider distributions of sarcomeres in mutant CMs would likely indicate adaptive changes to compensate the force generated by neighboring mature CMs.

5. In Figure 3G, what are the individual data points? The legend says n=9-12, but these individual data points do not seem to represent 9-12 different recordings.

: Each individual data point is a single CM. The n=9-12 means the total number of mice used to generate this data.

6. In Figure 4A the colors used for mutant and control are too similar. Can mutants use open symbols or symbols with a different shape.

: We agree that the colors similar, making it not easy to distinguish control and mutant cells. Unfortunately, this plot was created using Monocle R package software, which does not have the option to alter the shape. However, Figure 4b compares pseudotime and breaks it out into individual groups, so the maturation status and heterogeneity of control and mutant cells can be more clearly appreciated.

7. Figure 4F, the motif logos are displayed such that the letter height does not correspond to information content. This make the logos hard to interpret. In homer, use option -bits. Many nuclear receptors share motifs and it can be difficult to assign specific NRs to enriched motifs.

: We have substituted ChIP-seq data with ChIP-qPCR data as explained earlier.

8. In Figure 5, PGC1a agonists were used, but the authors show that PGC1 and PPAR levels are low in iPSC-CMs. Do these agonists upregulate PGC1 and PPAR? Otherwise it is hard to understand how activation of such lowly expressed genes induce the phenotypic changes. Are the changes abolished by PGC1 knockout or might these agonists have off-target activity?

: We chose to use the small molecules to activate PGC1/PPARa signaling for the two reasons: (1) Small PQQ is a potent PGC1 activator widely used in the field (Supruniuk et al., 2019; Chowandadisai et al., 2010; Kuo et al., 2015), and Wy14642 is a PPARa-specific ligand (Li et al., 2018). (2) PGC1 overexpression resulted in cellular hypertrophy (Fig. S4), and this phenotype is recapitulated by these small molecules (Fig. 5c). While PGC1/PPARa levels in PSC-CMs are similar to those in mid-late embryonic CMs (Fig. 2c), we have confirmed that PQQ treatment increases PGC1 readout genes and that Wy14643 treatment selectively increases expression of PPARa target genes, showing their specificity.

- Li G, Brocker CN, Xie C, et al. Hepatic peroxisome proliferator-activated receptor alpha mediates the major metabolic effects of Wy-14643. *J Gastroenterol Hepatol.* 2018;33(5):1138-1145. doi:10.1111/jgh.14046
- Chowanadisai W, Bauerly KA, Tchapanian E, Wong A, Cortopassi GA, Rucker RB. Pyrroloquinoline quinone stimulates mitochondrial biogenesis through cAMP response element-binding protein phosphorylation and increased PGC-1alpha expression. *J Biol Chem.* 2010;285(1):142-152. doi:10.1074/jbc.M109.030130
- Li Fang, et al. PPARgene: A Database of Experimentally Verified and Computationally Predicted PPAR Target Genes. *PPAR Research.* 2016: 6042162
- Krey, G. et al. Fatty acids, eicosanoids, and hypolipidemic agents identified as ligands of peroxisome proliferator-activated receptors by coactivator-dependent receptor ligand assay. *Mol Endocrinol.* 2017;11, 779-791, doi:10.1210/mend.11.6.0007

9. Line 244 references Figure 2E, which is absent.

: We have corrected the figure number (Fig. 5d).

Reviewer #4 (Remarks to the Author):

This manuscript reports additional information related to the role of the transcriptional coactivator PGC1a/b in cardiomyocyte maturation with implications in heart function in healthy and disease states. The authors developed and applied an experimental approach to perform large-particle sorting and analyzed RNA profiles in single myocytes from neonatal to adult stages. As expected, based on previous studies, they found that one of the major pathways associated with cardiomyocyte maturation is the transcriptional coactivator PGC1a/b and the downstream metabolic/mitochondrial network. The authors generated mosaic heart studies in vivo to delete PGC1a/b in a percentage of cardiomyocytes that were analyzed for gene expression analysis. Again, as predicted a series of genes linked to metabolism, calcium handling and size regulators were identified. A specific emphasis and analysis were pursued to YAP and SF3B2 splicing factor (it is known that PGC1a associates with splicing proteins) to define potential requirements for cell size. Overall, the manuscript is well executed and largely confirmed previous studies on PGC1a and heart biology and development. A major concern of this manuscript relates to 1) the novelty of the studies, and 2) incremental knowledge on PGC1a function in heart with the inclusion of YAP and SF32B, and 3) problematic use of pharmacological activators of PGC1a and PPARa.

1- Although the authors use an elegant series of experiments using scRNA seq in different cardiomyocytes population, the outcomes of these experiments are largely confirmative of previous studies.

: We respectfully disagree with this statement. Our study is conceptually novel and innovative for several reasons: (1) Previous PGC1/PPAR studies, largely focused on mitochondrial defects, manipulated the alleles in embryonic or adult stage, leaving their postnatal, cell-autonomous roles unclear. We found that neonatal mosaic deletion of PGC1 significantly downregulated genes involved in cardiac muscle development and impaired CM hypertrophy and contractility development, which has not been described previously. The underlying mechanisms are likely different from fatty acid oxidation (FAO) (Yang et al., 2019) because PGC1-deficient CMs do not

show no difference in genes and pathways involved in FAO. Instead, it regulates cardiac muscle development and oxidative phosphorylation (OXPHOS, Fig. 4e). (2) Using extensive scRNA-seq analysis, we discovered two novel targets (Yap1 and Sf3b2) of PGC1/PPARs mediating CM hypertrophy and contractility development, which is a completely new finding. This clearly shows context-dependent roles of PGC1/PPAR signaling. PGC1 itself has isoforms generated by splicing (Norrbon et al., 2011), but we were not able to find any literatures that PGC1 regulates splicing factors (please correct us if any). (3) We showed that PGC1/PPAR activity is low in PSC-CMs and has an instructive role in PSC-CM maturation. Furthermore, we demonstrated that Yap1 and Sf3b2 are required for PGC1/PPAR-mediated PSC-CM maturation. These are all novel findings that we expect to significantly advance the field.

- Yang, X. et al. Fatty acids enhance the maturation of cardiomyocytes derived from human pluripotent stem cells. *Stem Cell Rep.* 2019;13(4):657-668.
- Norrbom, J. et al. Alternative splice variant PGC-1a-b is strongly induced by exercise in human skeletal muscle. *Am J Physiol Endocrinol Metab.* 2011;301(6):E1092-8.

2- A major point the authors highlight is the cell size- mainly related to YAP- There are important caveats on these studies. First, manipulations of YAP in these studies by LOF and GOF are not sufficiently developed; but more importantly the connection between PGC1 and YAP is problematic- It is very likely that these effects are indirect through cell cell interactions created with the mosaic model. In addition, mTOR have been linked to PGC1 and also, differentiation or maturation in this case through PGC1a/b includes mitochondrial mass that might account for the cell size; or even the calcium handling pathways are associated with cell size.

: Thank you for the comment. While we cannot completely exclude the possibility of non-cell-autonomous effects in our mosaic model, this model has been well accepted and utilized widely in the field as an effective strategy to determine cell-autonomous function in vivo. The in vivo result is also supported by our in vitro experiment, where inhibiting Yap1 activity in PSC-CMs abolished cell growth mediated by PGC1/PPAR signaling (Fig. 5g). Together, these data suggest a required role of Yap1 in CM hypertrophy.

Our conclusion is supported by several papers published from the laboratories of Drs. Sean Sun and Jun Sadoshima (Perez-Gonzalez et al., 2019; Del Re et al., 2013). The paper from the Sun lab demonstrated that *Yap directly regulates cell volume via regulating cytoplasmic pressure*, and this occurs *independent of mTOR*. The paper from the Sadoshima lab showed that increased levels of Yap1 promotes hypertrophy in neonatal CMs. PGC1 was also shown to bind to the Yap1 promoter (Charos et al., 2012), suggesting a direct regulation.

- Perez-Gonzalez, N.A., et al. YAP and TAZ regulate cell volume. *J Cell Biol.* 218.10 (2019): 3472-3488.
- Del Re, D.P. et al. Yes-associated protein isoform 1 (Yap1) promotes cardiomyocyte survival and growth to protect against myocardial ischemic injury. *J Biol Chem.* 288(6): 3977-3988
- Charos, A.E. et al. A highly integrated and complex PPARGC1A transcription factor binding network in HepG2 cells. *Genome Res.* 22(9): 1668–1679.

3- Related to the previous point, the authors also include the SF3B2 as a novel mediator of PGC1a on cardiomyocyte maturation, through potential splicing. This could be a novel addition, but it will require large amount of experiments to characterize this factor.

: We believe the current data would be sufficient to draw the conclusion that SF3B2 is required for functional maturation of PSC-CMs driven by PGC1/PPAR signaling. To further strengthen the conclusion, we performed additional experiments and showed that SF3B2 is required for contractility development in addition to calcium handling development. We agree with the reviewer that it would be interesting to further investigate the mechanisms of SF3B2 mediating postnatal CM maturation. However, that is not the focus of the current manuscript and would be beyond the scope.

4- The use of PGC1a activator, is very, very problematic there is no experimental evidence that this is an specific activator of this protein and experiments with PGC1a/b mutants should define the specificity. Similar critiques apply to the 3PPARa-

: We chose to use the small molecules to activate PGC1/PPARa signaling for the two reasons: (1) PQQ is widely used as a potent PGC1 activator in the field (Supruniuk et al., 2019; Chowandadisai et al., 2010; Kuo et al., 2015), and Wy14642 is a PPARa-specific ligand (Li et al., 2018). (2) PGC1 overexpression resulted in cellular hypertrophy (Fig. S4), and this phenotype is recapitulated by the small molecules (Fig. 5c). Although our wet bench work capability is severely limited due to the COVID, we were able to confirm that PQQ/Wy14643 treatment increases expression of PGC1/PPARa readout genes, but not PPARg/d targets in PSC-CMs, showing their specificity.

- Supruniuk E, Miklosz A, Chabowski A, Łukaszuk B. Dose- and time-dependent alterations in lipid metabolism after pharmacological PGC-1 α activation in L6 myotubes. *J Cell Physiol.* 2019;234(7): 11923–11941
- Kuo Y.T., Shih P.H., Kao S.H., Yeh G.C., Lee H.M. Pyrroloquinoline Quinone Resists Denervation-Induced Skeletal Muscle Atrophy by Activating PGC-1 α and Integrating Mitochondrial Electron Transport Chain Complexes. *PLoS One.* 2015;10(12):e0143600
- Chowanadisai W, Bauerly KA, Tchapanian E, Wong A, Cortopassi GA, Rucker RB. Pyrroloquinoline quinone stimulates mitochondrial biogenesis through cAMP response element-binding protein phosphorylation and increased PGC-1 α expression. *J Biol Chem.* 2010;285(1):142-152.
- Li G, Bocker CN, Xie C, et al. Hepatic peroxisome proliferator-activated receptor α mediates the major metabolic effects of Wy-14643. *J Gastroenterol Hepatol.* 2018;33(5):1138-1145.
- Chowanadisai W, Bauerly KA, Tchapanian E, Wong A, Cortopassi GA, Rucker RB. Pyrroloquinoline quinone stimulates mitochondrial biogenesis through cAMP response element-binding protein phosphorylation and increased PGC-1 α expression. *J Biol Chem.* 2010;285(1):142-152.
- Li Fang, et al. PPARgene: A Database of Experimentally Verified and Computationally Predicted PPAR Target Genes. *PPAR Research.* 2016: 6042162
- Krey, G. *et al.* Fatty acids, eicosanoids, and hypolipidemic agents identified as ligands of peroxisome proliferator-activated receptors by coactivator-dependent receptor ligand assay. *Mol Endocrinol* **11**, 779-791.

5- The PPAR α is also problematic because it only defines a fraction of PGC1 α /b metabolic control linked to fatty acid oxidation, which is largely unexplored in this manuscript.

: We agree with the reviewer that PGC1 regulates numerous biological events including fatty acid oxidation, and PPAR α mediates a part of the events. However, the goal of this paper is to understand how PGC1/PPAR α signaling affects cellular hypertrophy and contractility development at the single cell transcriptomic and functional levels, not to investigate the mechanisms associated with the metabolic control.

REVIEWER COMMENTS

Reviewer #1 (Remarks to the Author):

While there are some limitations in the revision of the manuscript due to the equipment usage, the authors responded most of the reviewer's comments satisfactorily.

Reviewer #2 (Remarks to the Author):

This revised manuscript by Kwon and colleagues describes the role of PGC1/PPAR pathway to drive maturation of cardiomyocyte by regulating Yap1 and SF3B2. In response to the suggestions from the previous round of reviews, the authors have made significant efforts to improve the quality and reliability of this study's findings. In particular, the authors showed that single cardiomyocytes (CMs) from neonatal and adult mouse hearts exhibit wide variations of sizes and transcriptomes that continuously changed until adulthood. Additional loss-of-function studies show that PGC1 signaling mediated by Yap1 and SF3B2 is required for transcriptional maturation of CMs in vivo, revealing a previously unrecognized role of Yap1 and SF3B2 and demonstrating the power and utility of single cell transcriptional profiling of adult CMs. These studies are quite timely given the current interest in CM maturation and single cell transcriptome analysis and the identification of new role for Yap1 and SF3B2 are quite exciting. Remaining minor points that should be addressed include -

1. The authors' inclusion of a UMAP plot for the scRNAseq data is very much appreciated. I would further suggest to put the UMAP plot in Figure 1b instead of Supplemental Figure S1a and delete the t-SNE plot since there is really no need to have both t-SNE and UMAP plots of the same data in one manuscript.
2. Figure 2b – Perhaps the gene identity of the strongest hit (NFKBIA) should be labeled in the figure so that the readers would not look at this plot and wonder what the gene is. Also, an explanation of why this candidate was not chosen for further study as described in the response to Reviewer #1, Major Comments #1 should be added to the Results section.

Reviewer #3 (Remarks to the Author):

The authors addressed the points of my prior review, except that they did not perform the suggested experiments to assess the effect of PGC1 or PPARa agonist on t-tubules and conversion of energy substrate utilization from glucose to fatty acid. These are key features of cardiomyocyte maturation that iPSC-CMs lack. The authors also failed to address several key points raised by other reviewers, citing problems doing experiments due to the viral pandemic. While it is regrettable that the pandemic has delayed essential experiments, it should not change the standards of scientific review.

Reviewer #4 (Remarks to the Author):

The manuscript has improved after the revision. However, there are important points that the authors should include the limitations of their conclusions or remove the data from the manuscript. In some of the rebuttal points the answer is confused and not directly addressing the point. Please also refer to the initial critiques.

Previous Point- 4- The use of PGC1a activator.....because some articles have published it doesn't mean this is a PGC1a activator, these compounds are very promiscuous, and should not be included as PGC1a activators-

Previous point 5- The PPAR α is also problematic ...” However, the goal of this paper is to understand how PGC1/PPAR α signaling affects... the reply is confused as the only way to understand how a signal occurs is to provide mechanism.

RESPONSE TO CRITIQUES

Editor: “Thank you again for submitting your manuscript "PGC1/PPAR Drive Cardiomyocyte Maturation through Regulation of Yap1 and SF3B2" to Nature Communications. We have now received reports from 4 reviewers and, on the basis of their comments, we have decided to invite a revision of your work for further consideration in our journal. Your revision should address all the points raised by our reviewers (see their reports below). We ask that you perform the cardiomyocyte maturation experiments specified by Reviewer #3 and amend the text to clearly state the limitations of your approach as indicated by Reviewer #4.”

: We appreciate all the reviewers’ additional comments and the editor for inviting a revised manuscript. Our point-by-point responses to the reviewers’ comments are as follows:

Reviewer #1 (Remarks to the Author):

While there are some limitations in the revision of the manuscript due to the equipment usage, the authors responded most of the reviewer's comments satisfactorily.

: Thank you for the positive evaluation.

Reviewer #2 (Remarks to the Author):

This revised manuscript by Kwon and colleagues describes the role of PGC1/PPAR pathway to drive maturation of cardiomyocyte by regulating Yap1 and SF3B2. In response to the suggestions from the previous round of reviews, the authors have made significant efforts to improve the quality and reliability of this study’s findings. In particular, the authors showed that single cardiomyocytes (CMs) from neonatal and adult mouse hearts exhibit wide variations of sizes and transcriptomes that continuously changed until adulthood. Additional loss-of-function studies show that PGC1 signaling mediated by Yap1 and SF3B2 is required for transcriptional maturation of CMs in vivo, revealing a previously unrecognized role of Yap1 and SF3B2 and demonstrating the power and utility of single cell transcriptional profiling of adult CMs. These studies are quite timely given the current interest in CM maturation and single cell transcriptome analysis and the identification of new role for Yap1 and SF3B2 are quite exciting. Remaining minor points that should be addressed include -

1. The authors’ inclusion of a UMAP plot for the scRNAseq data is very much appreciated. I would further suggest to put the UMAP plot in Figure 1b instead of Supplemental Figure S1a and delete the t-SNE plot since there is really no need to have both t-SNE and UMAP plots of the same data in one manuscript.

: Thank you for the positive feedback. We agree, and have removed the tSNE plot.

2. Figure 2b – Perhaps the gene identity of the strongest hit (NFKBIA) should be labeled in the figure so that the readers would not look at this plot and wonder what the gene is. Also, an explanation of why this candidate was not chosen for further study as described in the response to Reviewer #1, Major Comments #1 should be added to the Results section.

: We have added this label to the graph and provided a rationale for not choosing this gene in the text.

Reviewer #3 (Remarks to the Author):

The authors addressed the points of my prior review, except that they did not perform the suggested experiments to assess the effect of PGC1 or PPAR α agonist on t-tubules and conversion of energy substrate utilization from glucose to fatty acid. These are key features of cardiomyocyte maturation that iPSC-CMs lack. The authors also failed to address several key points raised by other reviewers, citing problems doing experiments due to the viral pandemic. While it is regrettable that the pandemic has delayed essential experiments, it should not change the standards of scientific review.

: We have performed the following experiments to assess the effects of PGC1 and PPAR α agonist on t-tubule formation and conversion of energy substrate utilization:

To determine if PGC1 signaling affects t-tubule formation, we generated PGC1a/b DKO myocytes with a mT/mG reporter line in vivo¹. In this system, control and mutant cardiomyocytes express RFP and GFP in cell membrane, respectively, enabling us to examine t-tubules. Interestingly, t-tubules were detected in both control and mutant myocytes. This suggests that PGC1 signaling is not required for t-tubule formation. In fact, the mechanisms underlying t-tubule morphogenesis is not well understood, and our observation suggests that the morphogenetic events are not adversely affected by the lack of PGC1 signaling. On a related note, a recent paper demonstrated the necessity of 3D environment in generating t-tubule-like structure in PSC-CMs², suggesting that t-tubule formation could be an adaptive process in response to microenvironmental changes. While we acknowledge t-tubule formation is an important feature of mature cardiomyocytes, we still believe our conclusions holds on the role of PGC1/PPAR in cellular hypertrophy and contractility development.

Next, to examine whether PGC1 and PPAR agonists mediate a switch from glycolysis to fatty acid, we quantified the ratio of oxygen consumption rate (OCR, an indicator of mitochondrial respiration) to extracellular acidification rate (ECAR, an indicator of glycolysis) with the Agilent Seahorse XFe96 Analyzer. We found that the treatment significantly lowers the ratio, suggesting a conversion from glucose to fatty acid in substrate utilization. Consistently, the treatment led to downregulation of key glucose utilization genes (*PKM2*, *SLC16A3*, *SLC2A1*, *TP11*).

Reviewer #4 (Remarks to the Author):

The manuscript has improved after the revision. However, there are important points that the authors should include the limitations of their conclusions or remove the data from the manuscript. In some of the rebuttal points the answer is confused and not directly addressing the point. Please also refer to the initial critiques. Previous Point- 4- The use of PGC1a activator.....because some articles have published it doesn't mean this is a PGC1a activator, these compounds are very promiscuous, and should not be included as PGC1a activators-

: Following Reviewer #4's suggestion, we changed the text to "a small molecule that is shown to activate PGC1 signaling" from "PGC1 activator" to further clarify that the use of PQQ. In addition, we added this sentence in Discussion "It is also worth pointing out that, while we used PQQ to pharmacologically activate PGC1 signaling, PQQ, as a redox cofactor, may also have other biological effects on mammalian cells that have not been fully characterized."

Previous point 5- The PPARa is also problematic ..." However, the goal of this paper is to understand how PGC1/PPARa signaling affects... the reply is confused as the only way to understand how a signal occurs is to provide mechanism.

: We apologize for our misunderstanding and the confusion. We acknowledge that future work would be certainly needed to understand how PGC1/PPARa regulate CM maturation via YAP1/SF3B2. We added this in Discussion.

REVIEWERS' COMMENTS

Reviewer #3 (Remarks to the Author):

The authors performed two additional experiments as suggested by this reviewer. First, the authors showed that PGC1a/b DKO myocytes have qualitatively normal T-tubules. This suggests that PGC1a/b is required for some but not all aspects of CM maturation. The authors should include these data with sufficient replicates in their manuscript and not as a figure for the reviewer. This result should lead to modification of the conclusion that PGC1a/b is a master regulator of CM maturation. Rather it regulates some but not all aspects of CM maturation. This does not diminish the importance of this study but refines the conclusion about the role of PGC1a/b in maturation.

Second, the authors showed that ECAR/OCAR ratio decreased in cells (? iPSC-CMs -- cells were not identified) treated with the PGC1 and PPAR1 agonists. A more informative way to plot these data is with ECAR on the Y axis and OCR on the X axis. This will provide information as to whether oxygen consumption increases, extracellular acidification decreases, or both. The authors should not equate fatty acid metabolism with changes in OCR, especially if the culture conditions did not alter the availability of fatty acids in the media. The authors should also include these data in the manuscript rather than present them as a figure for the reviewer.

RESPONSE TO CRITIQUES

Reviewer #3 (Remarks to the Author):

The authors performed two additional experiments as suggested by this reviewer. First, the authors showed that PGC1a/b DKO myocytes have qualitatively normal T-tubules. This suggests that PGC1a/b is required for some but not all aspects of CM maturation. The authors should include these data with sufficient replicates in their manuscript and not as a figure for the reviewer. This result should lead to modification of the conclusion that PGC1a/b is a master regulator of CM maturation. Rather it regulates some but not all aspects of CM maturation. This does not diminish the importance of this study but refines the conclusion about the role of PGC1a/b in maturation.

: We included the data in Figure S2o and modified the text as suggested.

Second, the authors showed that ECAR/OCAR ratio decreased in cells (? iPSC-CMs -- cells were not identified) treated with the PGC1 and PPAR1 agonists. A more informative way to plot these data is with ECAR on the Y axis and OCR on the X axis. This will provide information as to whether oxygen consumption increases, extracellular acidification decreases, or both. The authors should not equate fatty acid metabolism with changes in OCR, especially if the culture conditions did not alter the availability of fatty acids in the media. The authors should also include these data in the manuscript rather than present them as a figure for the reviewer.

: We included this data in Figure S3h as suggested.